Li *et al. Genome Biology*      (2023) 24:70

**METHOD**

# FIPRESCI: droplet microfluidics based combinatorial indexing for massive-scale 5′-end single-cell RNA sequencing

Yun Li[1,2,3†], Zheng Huang[1,2,3†], Zhaojun Zhang[1,2,3,4†], Qifei Wang[1,2,3], Fengxian Li[5], Shufang Wang[5], Xin Ji[6], Shaokun Shu[7], Xiangdong Fang[1,2,3,8,9] and Lan Jiang[1,2,3,4,9,10*]

†Yun Li, Zheng Huang, and Zhaojun Zhang contributed equally.

*Correspondence:
jiangl@big.ac.cn

[1] China National Center for Bioinformation, Beijing 100101, China
Full list of author information is available at the end of the article

## Abstract

Single-cell RNA sequencing methods focusing on the 5′-end of transcripts can reveal promoter and enhancer activity and efficiently profile immune receptor repertoire. However, ultra-high-throughput 5′-end single-cell RNA sequencing methods have not been described. We introduce FIPRESCI, 5′-end single-cell combinatorial indexing RNA-Seq, enabling massive sample multiplexing and increasing the throughput of the droplet microfluidics system by over tenfold. We demonstrate FIPRESCI enables the generation of approximately 100,000 single-cell transcriptomes from E10.5 whole mouse embryos in a single-channel experiment, and simultaneous identification of subpopulation differences and T cell receptor signatures of peripheral blood T cells from 12 cancer patients.

**Keywords:** Combinatorial indexing, Sample multiplexing, Single-cell RNA-seq, Single-nucleus RNA-seq, scVDJ-seq, scTCR-seq, 10X Genomics

## Background

Droplet microfluidic-based 5′-end or 3′-end single-cell and single-nucleus RNA sequencing (scRNA-seq and snRNA-seq) have emerged as central tools for interrogating the cellular states of whole organs and entire organisms. While with similar capability for cell typing, 5′-end based methods offer several advantages over 3′-end. First, 5′-end single-cell transcriptomics can detect the location of cis-regulatory elements (CREs; including promoters and enhancers) and quantify their in vivo activity [1]. Cell atlas studies that rely on 5′-end sc/snRNA-seq to infer cis-gene regulatory network in the human body have been launched [2]. Second, the 3′-end method is difficult to identify full-length immune receptor (TCR and BCR) sequences in a typical short-read sequencing setting [3]. 5′-end scRNA-seq coupled with single-cell V(D)J sequencing has become the most popular approach to reveal the clone type-specific transcriptional signatures

which is a challenge for conventional bulk immune receptor sequencing. The procedure has been widely applied to the study of antibody–drug discovery [4] and characterizes the immune response in patients of COVID-19 [4, 5], cancer, autoimmune diseases, and neurodegenerative diseases [6]. Third, for snRNA-seq, which is critical for profiling tissues that are hard to dissociate or frozen clinical samples, the 5′-end method provides a unique opportunity to improve the transcriptome complexity through reverse transcription by customized oligo priming. Furthermore, by introducing a primer that targets a conserved region of bacterial 16S rRNA, the 5′-end method enables to co-detect the presence of intratumoral bacteria and transcriptome in the same single tumor cell simultaneously [7]. However, only a limited number of protocols have been developed to capture the 5′-end of transcripts [8]. The existing droplet-based 5′-end single-cell transcriptomics approaches are costly and preclude large-scale studies on millions of cells or thousands of samples.

The droplet microfluidic system uses an ineffective way to minimize the doublet rate in single-cell analysis, which is constrained by a Poisson-like distribution. The single-cell suspension is loaded into the microfluidic device at very low concentrations (typically 500–8000 cells), making it unlikely that two cells enter the same droplet (~ 100,000 droplets per channel for Chromium Controller, ~ 200,000 droplets for Chromium X). Sample multiplexing has been developed to overcome this throughput limitation by identification and removal of doublets and achieve experimental cost-reducing and batch effects removal by allowing overloading. The strategies are initially introduced by using natural genetic variations and expanded by oligo-tagged antibodies (Cell hashing) [9], lipid-tagged indices [10], lentiviral-based CellTag Indexing [11], methyltetrazine-modified DNA oligonucleotides [12], and so on. However, those strategies only increase the cell throughput modestly as the data of doublets are not resolved and need to be discarded after being sequenced.

Conceptually different approaches that use droplet microfluidic to accomplish single-cell combinatorial indexing have emerged recently. DsciATAC-seq [13], scifi-RNA-seq [14], and SCITO-seq [15] have achieved ultra-high-throughput analysis for chromatin accessibility, 3′-end gene expression, and surface protein expression, respectively. The target molecules (open chromatin DNA, transcripts, antibody-derived tags) are first labeled by a well-specific barcode and subsequently by a large number of droplet-specific barcodes in a two-round barcoding experiment. The droplet is not considered as a way to isolate single-cell but a physical compartment that is conceptually equivalent to a "well" in split-pool single-cell combinatorial indexing protocols, such as sci-RNA-seq [16] and SPLiT-seq [17]. Compared with the sample multiplexing-based doublets removing methods, those strategies can substantially increase the cell throughput as the data of doublets will be resolved rather than discarded. Compared with plate-based single-cell combinatorial indexing laboratory protocols, which typically need 3- or 4-round barcoding, those strategies are much easier to perform by leveraging the powerful indexing ability of droplet microfluidic for the second round. However, the droplet microfluidic-based combinatorial indexing 5′-end single-cell RNA-seq has not been described.

Here, we developed FIve PRime End Single-cell Combinatorial Indexing RNA sequencing (FIPRESCI), a simple and highly efficient workflow for 5′-end scRNA-seq and snRNA-seq. Our approach combines preindexing of the whole transcriptomes

in situ through Tn5 transposome with droplet template switch oligo (TSO) barcoding in a commercial microfluidic platform. We demonstrate the signal from FIPRESCI sc(n) RNA-seq colocalizes with cis-regulatory elements detected from ATAC-seq. We further demonstrate FIPRESCI sc(n)RNA-seq can increase the cell throughput by at least tenfold over the existing standard procedure and resolve the cellular diversity of whole mouse E10.5 embryo. Finally, we demonstrate the ability of FIPRESCI for sample multiplexing and multimodal profiling of the whole transcriptome and immune receptor repertoire.

## Results

### FIPRESCI overview

Previous studies have shown that the Tn5 transposase enzyme maintains the contiguity of target dsDNA, and the protein-DNA complex is only dissociated after the addition of a protein-denaturing agent such as SDS [18]. Recently, the Tn5 transposome has been reported to also possess in vitro tagmentation activity towards both strands of RNA/DNA hybrids which are typical products of reverse transcription step during RNA-seq experiment [19, 20]. Motivated by these observations, we asked whether or not the Tn5 transposase also stayed bound to its RNA/cDNA substrate after transposition. The results indicate that each RNA/cDNA molecule subjected to transposition retained its high molecular weight and was only fragmented when exposed to SDS (Additional file 1: Fig. S1a). We, therefore, postulated that in a condition without a protein-denaturing agent, indexed Tn5 transposome could be used to preindex whole transcriptomes at a single molecular level after reverse transcription inside single nuclei or permeabilized cells. The in situ reverse transcriptions followed by a Tn5 tagmentation step did not change the morphological property of nuclei or permeabilized cells (Additional file 1: Fig. S1b). We further confirmed that the Tn5 barcoded 5′-end transcript can be captured by TSO-containing barcoding primers used in microfluidic droplet single-cell sequencing. The success of highly efficient two-round combinatorial indexing of FIPRESCI is built on the key insight that droplet microfluidic can tolerate substantial overloading while maintaining a low collision rate when single molecular preindexing is available to resolve transcriptome of different cells/nuclei within the same droplet.

The workflow of FIPRESCI is described in Fig. 1a (see also Additional file 1: Fig. S2). Briefly, (i) prepare permeabilized cells or nuclei; (ii) fixed and permeabilized cells or nuclei are reverse transcribed using an Oligo d(T) primer or random primer; (iii) permeabilized cells or nuclei are distributed in multiwell plates (we use one 96-well) to be labeled with well-specific preindexing (round1) barcodes by indexed Tn5 transposase to mark cDNA; (iv) the cells or nuclei with preindexed cDNA are pooled, randomly mixed, and encapsulated into emulsion droplets with a high degree of overloading. Inside these overloaded droplets, the 3′-end of cDNA which was added with 2–5 untemplated nucleotides by the Moloney murine leukemia (M-MLV) reverse transcriptase, are labeled with oligonucleotides containing droplet-specific barcode (round2), UMI, and template switch oligo (TSO) by template-switching and cDNA extension. Such, the round1 barcodes are shared between all cells from the same well, the round2 barcodes are shared between all cells in the same droplet, but the combination of the two barcodes uniquely identifies transcripts derived from the same single cell; (v) the droplets are broken and

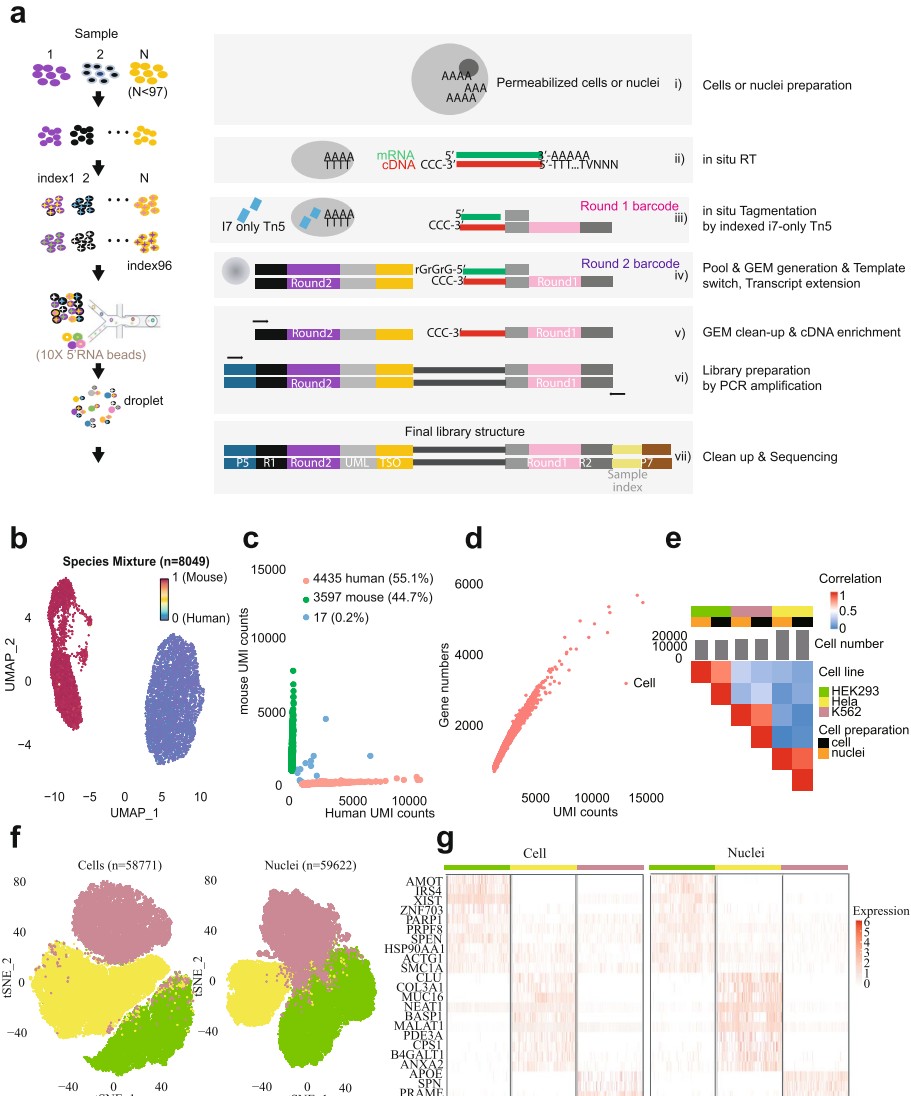

**Fig. 1** Overview and validation of FIPRESCI. **a** The FIPRESCI schematic workflow and detailed method design. Permeabilized cells or nuclei are reverse transcribed, then nuclei or cells are randomly distributed into wells containing indexed Tn5 transposome to label the cellular origin of RNA/cDNA hybrid heteroduplexes within cells. The cells or nuclei containing preindexed cDNA are pooled, randomly mixed, and encapsulated using a commercial microfluidic platform and amplified for preparation of the sequencing library. **b** Species-mixing experiment with a library prepared from the 1:1 mix of human (Jurkat) and mouse (NIH-3T3) permeabilized cells. Human uniquely barcoded cells (UBCs) are blue, mouse UBCs are red in UMAP. *n* = 8049 cells. **c** The number of unique fragments aligning to the human or mouse genome. Human UBCs are red, mouse UBCs are green, and mixed-species UBCs are blue. The estimated barcode collision rate is 0.2%, whereas species purity is > 99%. **d** The number of UMI counts plotted against detected genes from species-mixing experiments. **e** Heatmap showing pairwise correlations and hierarchical clustering for the gene expression profiles across cell lines, cell preparation methods using FIPRESCI. **f** Dimensionality reduction (UMAP) and unsupervised clustering for single-cell (*n* = 58,771) and single-nucleus (*n* = 59,622) FIPRESCI of the three cell lines. HEK293 is red, Hela is green, and K562 is blue. **g** Heatmap showing differentially expressed genes and gene expression levels of single-cell and single-nucleus FIPRESCI for three cell lines. Each column represents a single cell

pooled after template switch reaction mixtures are recovered; (vi–vii) cDNA labeled with round1 and round2 are enriched by PCR amplification and prepared for sequencing; (viii) (optional) for immune cells, T cell receptor (TCR), or B-cell immunoglobulin

(Ig) transcripts can be enriched from the cDNA products via PCR amplification with primers specific to either the TCR or Ig constant regions to measure immune repertoire information.

## FIPRESCI validation

As a proof of concept, we performed FIPRESCI on a mixture of human (Jurkat) and mouse (NIH 3T3) cell lines. During the first round of indexing, 96 wells contained 1:1 mixed human and mouse cells, and then 15,300 cells which is the maximum recommended loading concentration were loaded to generate emulsion droplets by the 10X Genomics Chromium system. We recovered 8049 high-quality (UMI > 1000) single-cell transcriptomes and readily assigned cells as human or mouse (Fig. 1b). Notably, nearly all (99.8%) of the uniquely barcoded cells were unambiguously assigned to a single species (> 90% of reads aligned to a single genome), only 17 cells representing barcode collisions between mouse and human cells were identified, representing a remarkably low 0.2% collision rate, which is much lower than expected multiplet rate of 7.6% when the same number of cells are loaded in the standard 10X Genomics procedure (Fig. 1c, and Additional file 1: Fig. S3a). Figure 1d shows the number of UMI counts plotted against detected genes from species-mixing experiments.

   After assessing technical feasibility, we sought to benchmark the scalability and performance of FIPRESCI by performing scRNA-seq and snRNA-seq using 3 human cell lines (HEK293T, Jurkat, and K562) with variable cell size and mRNA content. Permeabilized cells and nuclei were processed with FIPRESCI workflow respectively. We marked technical replicates of each cell line with different round1 barcodes by seeding each cell line to 32 wells of a 96-well plate, then in both cases loading 100,000 cells/nuclei per microfluidic channel (Additional file 1: Fig. S3b). We recovered transcriptome from 58,771 cells and 59,622 nuclei, with a recovery rate of over 58%. We also calculated the effective number of individual nuclei or cells (as identified by round1 barcodes) in each droplet (as identified by round2 barcodes) from the sequencing data. Most droplets (96.38% for scRNA-seq, 98.25% for snRNA-seq) contain 1–3 cells (Additional file 1: Fig. S3c-d). After dimension reduction using uniform manifold approximation and projection (UMAP), all three cell lines were separated by their transcriptomes in both permeabilized cells and nuclei (Fig. 1f). The recovery rate of these cell lines is similar, suggesting it is cell size and mRNA content independent. Differential expression of marker genes validated these cluster identities. In addition, the marker genes of three cell lines were consistently highly expressed in the permeabilized cells and nuclei data (Fig. 1g). Estimates of gene expression from the aggregated transcriptomes of nuclei and cells were well correlated [Spearman correlation coefficient ($r$) = 0.81 for HEK293T, 0.89 for K562, and 0.85 for Hela; Fig. 1e]. These experiments confirm that FIPRESCI is robust across different input materials and scales well while maintaining a high recovery rate.

## Optimization of FIPRESCI

We attempted to optimize the performance of FIPRESCI so that the methods are improved with sensitivity and compatible with a broader range of samples. We first try to find robust conditions for an efficient tagmentation on RNA/DNA hybrid heteroduplexes within permeabilized cells or nuclei. We focused on the Hela cell line and

performed a set of experiments with different reaction buffers and variations according to the previously published studies [18–21] and three available commercial tagmentation buffers (the detailed components of these tagmentation buffers and the design of the experiments are provided in Additional file 1: Fig. S4a.c and Supplementary Table 3). Commercial buffer 1 showed the best performance. In addition, Tris-based or TAPS-based buffer only containing dimethylformamide (DMF) also showed the efficient tagmentation of FIPRESCI at matched sequencing depths compared with other custom buffers (Fig. 2a, and Additional file 1: Fig. S4b). Therefore, for the rest of the subsequent experiments, we used the customized Tris-based DMF tagmentation buffer (10 mM Tris–HCl at pH 7.5, 5 mM $MgCl_2$, and 10% DMF). Unexpectedly, in contrast to the results in tagmentation activity on free RNA/DNA hybrid duplexes [19], we found the addition of crowding agents such as PEG 8000 and PEG200 showed poor performance (merely 10–30% lower average genes per cell detected compared with the buffers containing DMF only) on FIPRESCI. We speculate that since in FIPRESCI transposition reaction occurs inside intact nuclei or cells, the presence of crowding agents may hinder the penetration of Tn5 or magnesium ions into the cells/nuclei.

Next, we assessed the performance of different RT primers, including oligo d(T), random hexamer primer, and equimolar concentration mix of oligo d(T) and random hexamer primer on FIPRESCI for both permeabilized Hela cells and nuclei. Each condition was marked with 8 different round1 barcodes as technical replicates, (the design of the experiments is provided in Additional file 1: Fig. S5a) then all the cells/ nuclei were pooled and loaded into a single microfluidic channel of the Chromium system. After sequencing, we assessed the sensitivity of gene detection in different RT primer methods. At matched sequencing depths, the FIPRESCI for single cell using oligo d(T) primer methods always detected more genes than other primer conditions (Fig. 2b). As expected, oligo d(T) primer methods in single-nucleus FIPRESCI data detected fewer genes per nucleus compared to FIPRESCI using cells. However, single-nucleus FIPRESCI using mixed RT primers identified more genes than the oligo d(T) primers, and the random primer methods. The sensitivity of gene detection in mixed RT primer was better than the random RT primer for FIPRESCI using nuclei. Thus, mixed RT primer methods could be an attractive strategy to improve the mRNA recovery for single-nucleus RNA-seq. Notably, at a typical sequencing depth of 50 k fragments per cell, we obtain a median of 2239 and a mean of 2150 genes per cell with oligo d(T) primer(Fig. 2b), which is comparable with 10X Genomics standard 5′-end scRNA-seq procedure.

In addition, the reads from each method are strongly enriched at the 5′-end of genes and peaked exactly at the annotated transcription start sites (TSSs), suggesting our methods can accurately identify active TSSs (Fig. 2c and Additional file 1: Fig. S5b (upper)). Then, we evaluated the sensitivity of enhancers detection in various RT primer methods data by calculating the coverage of reads around the enhancer center. Enhancers were defined as distal ATAC peaks whose distance to the nearest gene TSS is greater than 2000 bp. Notably, in all RT primer conditions, there is a bimodal distribution centered around the start and end site of the distal ATAC peak (Fig. 2d). FIPRESCI using nuclei methods showed higher sensitivity, especially by using a random RT primer. If an ATAC-defined enhancer region is enriched with a strong FIPERSCI-seq signal (RPKM > 1 and adjusted *p*-value < 0.01, see "Methods"), we defined it as an eRNA locus. About 2% of

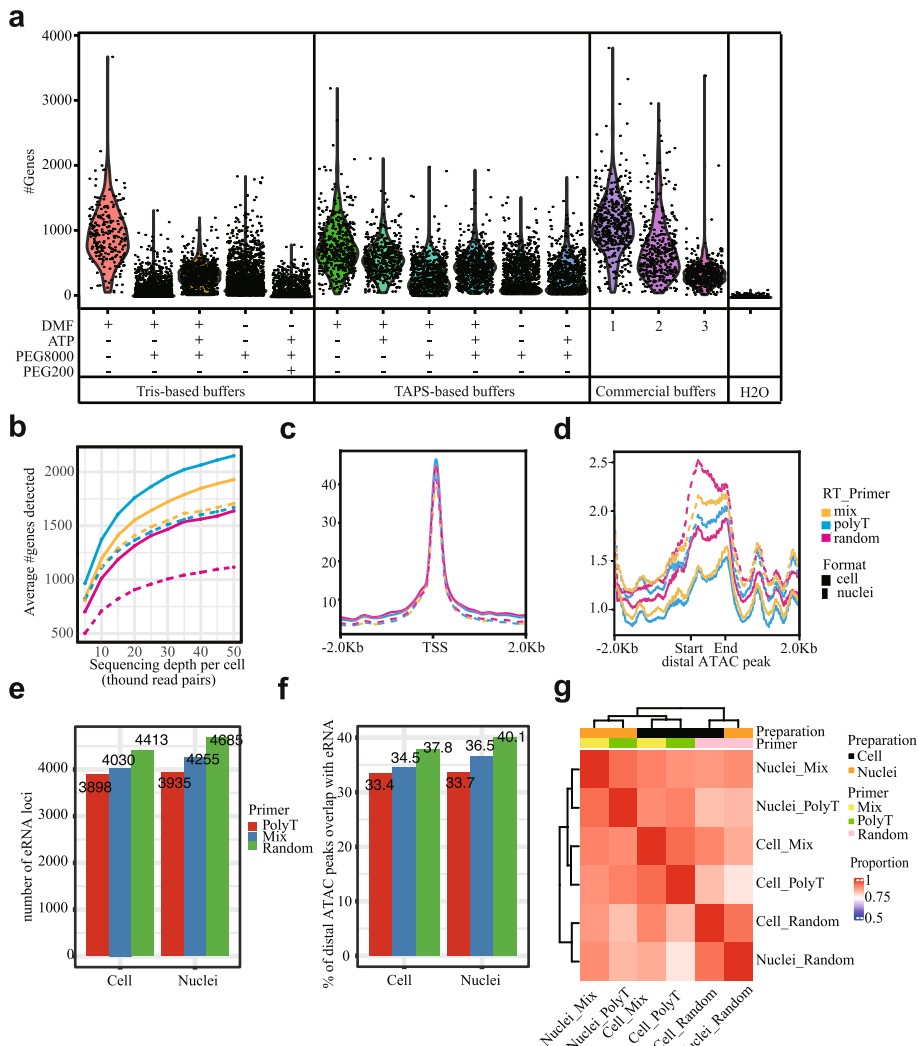

**Fig. 2** Optimization and expanded application of FIPRESCI. **a** Violin plot showing sensitivity in FIPRESCI generated with a set of TN5 tagmentation buffers. Each dot represents a single cell. *Y*-axis is the number of genes detected. **b** Sensitivity of gene detection in three reverse transcription primer conditions in FIPRESCI single-cell RNA-seq (solid line) and single-nucleus RNA-seq (dotted line) across sequencing depth. **c** Distribution of reads from three reverse transcription primer conditions in FIPRESCI scRNA-seq and snRNA-seq around annotated TSS. **d** Distribution of reads from three reverse transcription primer conditions in FIPRESCI scRNA-seq and snRNA-seq around enhancer center. **e** Histogram showing the number of eRNA loci identified in three reverse transcription primer conditions in FIPRESCI scRNA-seq and snRNA-seq. **f** Histogram showing the percentage of distal ATAC peaks which are overlap with eRNA loci in three reverse transcription primer conditions in FIPRESCI scRNA-seq and snRNA-seq. **g** Heatmap showing pairwise correlations, hierarchical clustering for the eRNA profile across different reverse transcription primers, and preparation methods using FIPRESCI

the reads are generated from eRNA loci(Additional file 1: Fig. S5c). We found FIPERSCI can identify 3898~4685 eRNA loci, which account for 33.4~40.1% distal ATAC peaks (Fig. 2e,f). Furthermore, the eRNA profile among different conditions is similar, which indicated those eRNAs may be derived from strong enhancers (Fig. 2g). Interestingly, we noticed that the rRNA detected in FIPRESCI are unexpectedly low, which is similar to

random primer RT-based protocol SUPeR-seq [22]. The cell lysis buffer which lacks Proteinase K may contribute to the low rRNA level (Additional file 1: Fig. S5d,e,f).

Recent technological advances have made it possible to jointly profile transcriptome and chromatin accessibility within the same single cell. However, the widespread application of single-cell co-assay is challenging due to its experimental complexity, scalability, and cost [23]. Since FIPRESCI has achieved ultra-high-throughput sc(sn)-RNA-seq and detected enhancers, we further test its feasibility in computationally predicting ATAC-seq data from RNA-seq. We modified BABEL [23] to build a deep learning model that can predict scATAC-seq signals from the Hela cells FIPRESCI dataset. We observed that adding enhancer RNA (eRNA) information during training, which is defined as a distal FIPRESCI signal, outperforms the standard models that only include mRNA (gene expression level) (Additional file 1: Fig. S5g). The correlation of aggregated predicted ATAC peak signals with bulk ATAC-seq peak signals is 0.33 (restrict to top 6000 peaks) (Additional file 1: Fig. S5h). Finally, we found the predicted ATAC signal based on FIPRESCI data recapitulates the actual ATAC-seq signals (Additional file 1: Fig. S5i).

In summary, the results demonstrate that FIPRESCI can detect gene expression and identify cis-regulatory elements in the same cell/nucleus. The overall performance of different sample formats and RT primer conditions are similar, while scRNA-seq is suitable to detect more genes and snRNA-seq has the advantage to detect more enhancers.

### 5′-end snRNA-seq of mouse whole embryo in a single 10X Genomics channel

To test whether FIPRESCI is feasible for various primary tissue and demonstrates scalability, we performed FIPRESCI snRNA-seq on a mouse E10.5 whole embryo. We collected 2 C57BL/6N × PWK/PhJ hybrid mouse embryos at E10.5. The nuclei isolated from each embryo were pooled, counted, and evenly deposited in 96-well plates with an expectation of 4000 nuclei per well. The first round index introduced by Tn5 does not contain sample identity but will exponentially increase the cell throughput when combined with the second round barcode from droplet microfluidics. We recovered transcriptomes of 117,804 nuclei from a single 10X Genomics channel, which is over tenfold compared to the standard procedure in terms of cell throughput.

Initially, analysis by unsupervised clustering identified 15 well-separated clusters (Fig. S6a, left). Multiple distinct cluster-specific marker genes can be found for each cluster (Fig. S6a, right), indicating the high quality and complexity of the data. We further annotated each cluster by integration analysis with a reference dataset [24]. Encouragingly, the majority of the cells ($n = 101{,}031$, 85.8%) can be assigned a reliable cell type label (Fig. 3a, left and Additional file 1: Fig. S6b-c). In addition, the proportion of each cell type in FIPRESCI is highly consistent with the counterpart in the reference E10.5 embryo dataset (Fig. 3b). What is more, the FIPRESCI captured the 5′-end of the transcripts allowing us to investigate the landscape of TSSs usage in a cell type and single-cell specific manner. For example, Specc1 uses the most upstream TSS in cluster 7 and cluster 9 while using a downstream TSS that skips the first three exons in most of the rest of the clusters (Fig. 3c).

A previous study has reported that with the brain-related samples, relative TSS usage significantly varied concerning upstream or downstream position within each gene [8]. Next, we set out to ask whether FIPRESCI can identify TSS switches during brain

development and GABAergic neurogenesis in the E10.5 mouse embryo. First, we isolate the brain cells from the whole embryo dataset and perform a pseudo time analysis. Multiple gene modules which are highly correlated with the trajectory graph are identified (Fig. 3e, left, and Additional file 1: Fig. S6d-g). The cell type labels along the inhibitory neuron trajectory confirmed that the neural tube and radial glia cells are more enriched in the early, neural progenitor cells are enriched in the intermediate, and inhibitory neuron progenitors and inhibitory neurons are enriched at late stages (Additional file 1: Fig. S6h). Five-gene group dynamic pattern is identified along the pseudo time and mainly include early active (group 4), intermediate active (group 5), and late active (groups 1 and 2) (Fig. 3e, right). What is more, we aggregate the FIPRESCI signal from the cells of inhibitory neuron lineages based on the pseudo time into three stages and identify several dozens of TSS switches from the paired comparison of adjacent stages (Fig. 3f, up). A representative example is Rbfox2 (Fig. 3f, bottom), also known as RBM9, a multifunction RNA binding protein, which has a well-established role in alternative splicing [25] and recruitment of PRC2 [26]. Our results clearly show that at the early stage, Rbfox2 mainly uses the most upstream promoter while the downstream promoter which skips the first two exons get strongly activated at the intermediate and late stages (Fig. 3f, bottom). Our results provide a resource for future studies to dissect the function and mechanism of the cell type-specific and developmental regulated TSS switch. Collectively, those results suggest FIPRESCI reveals the heterogeneity of complex cellular systems with high efficiency and flexibility and uncover widespread previously unexplored TSS switch (Additional file 1: Fig. S6i).

### FIPRESCI enables efficient single-cell transcriptional profiling of human peripheral T cells

To further prove that our method can be used for large-scale single-cell atlas projects by substantially increasing the sample throughput by multiplexing, we applied FIPRESCI to human primary T cells from 14 donors across 5 cancer types and healthy people. Each cancer type contains at least 2 donors. CD3+T cells are sorted from PBMC by FACS. All the samples are processed in a single FIPRESCI experiment so the batch effects will be minimized. We use 96-plex in the first round of indexing, thus 6–12 different (round1) barcodes may represent the same sample identity and the corresponding data can be considered as technical replicates. In parallel, we enriched the V(D)J sequences from transcriptome libraries by PCR for single-cell T cell receptor(TCR) mapping (Fig. 4a, left and middle).

(See figure on next page.)
**Fig. 3** FIPRESCI enables analysis of E10.5 whole mouse embryo. **a** Label transfer from a scRNA-seq profiled mouse organogenesis atlas to FIPRESCI data. UMAP embedding colored by transferred cell types. **b** A high correlation between cell type proportion was detected by Fipresci-Seq and the public atlas. **c** Differential usages of TSS for gene Specc1. IGV (Integrative Genomics Viewer) Track plot shows cluster 7 and cluster 9 use the same TSSs of gene Specc1, while other clusters used different ones. The tracks are group scaled and the range is shown on the right margin. **d** De novo constructing inhibitory neuron trajectory. **e** Trajectory graph-correlated genes. UMAP plots colored by gene expression (left), those genes are highly correlated with the brain trajectory graph. Heatmap (right) showing genes dynamic changes along with pseudo time within inhibitory neuron trajectory. **f** Top, heatmap showing TSS usage proportion over 3 stages of inhibitory neuron trajectory. Row names of the heatmap are stages (early, medium, and later pseudo time), the column represents TSSs, and colors indicate TSS reads proportion of all TSS reads in the corresponding gene within one stage (early, medium, or later cells). Track plot (bottom) showing gene Rbfox2 TSS usage changes along the inhibitory trajectory

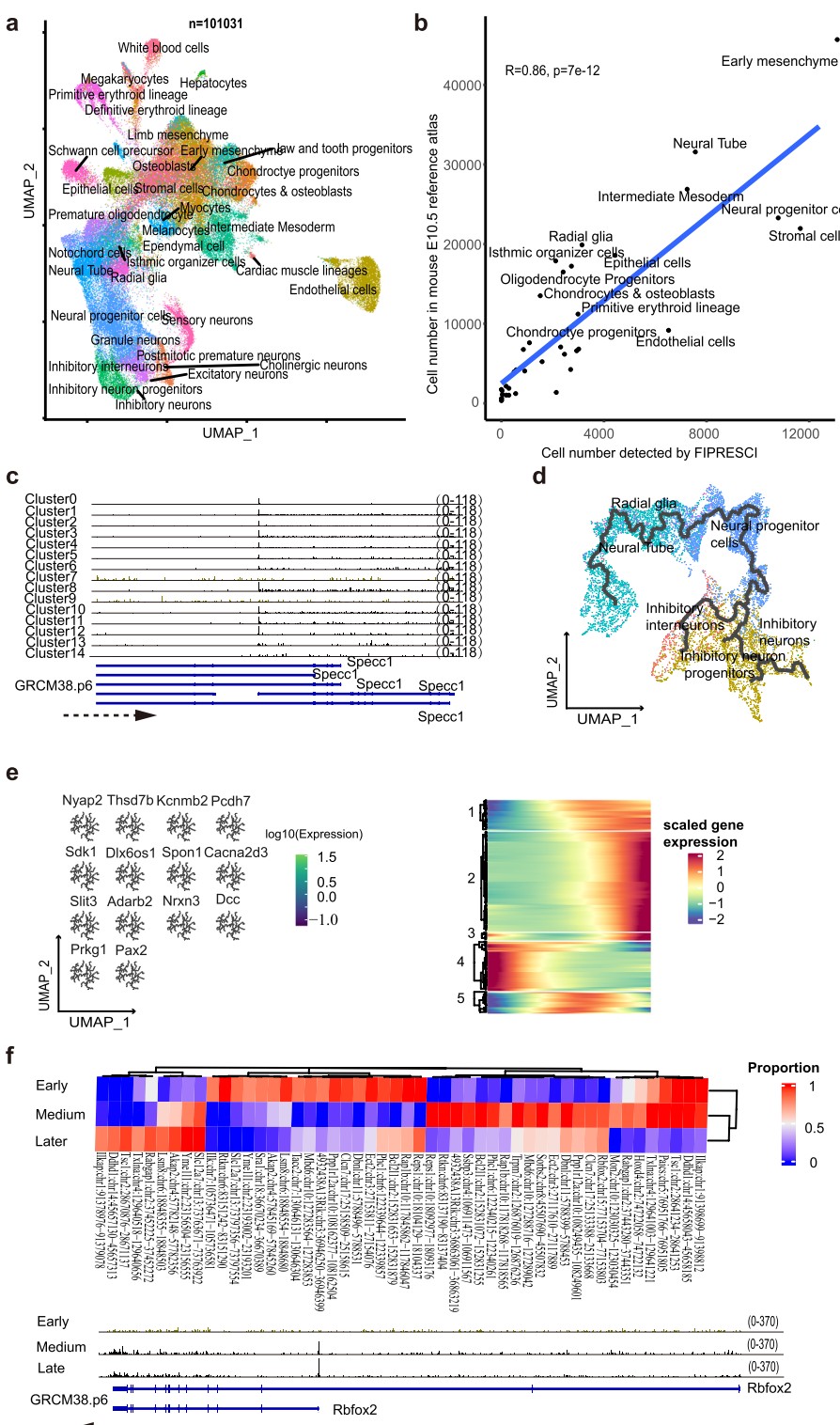

**Fig. 3** (See legend on previous page.)

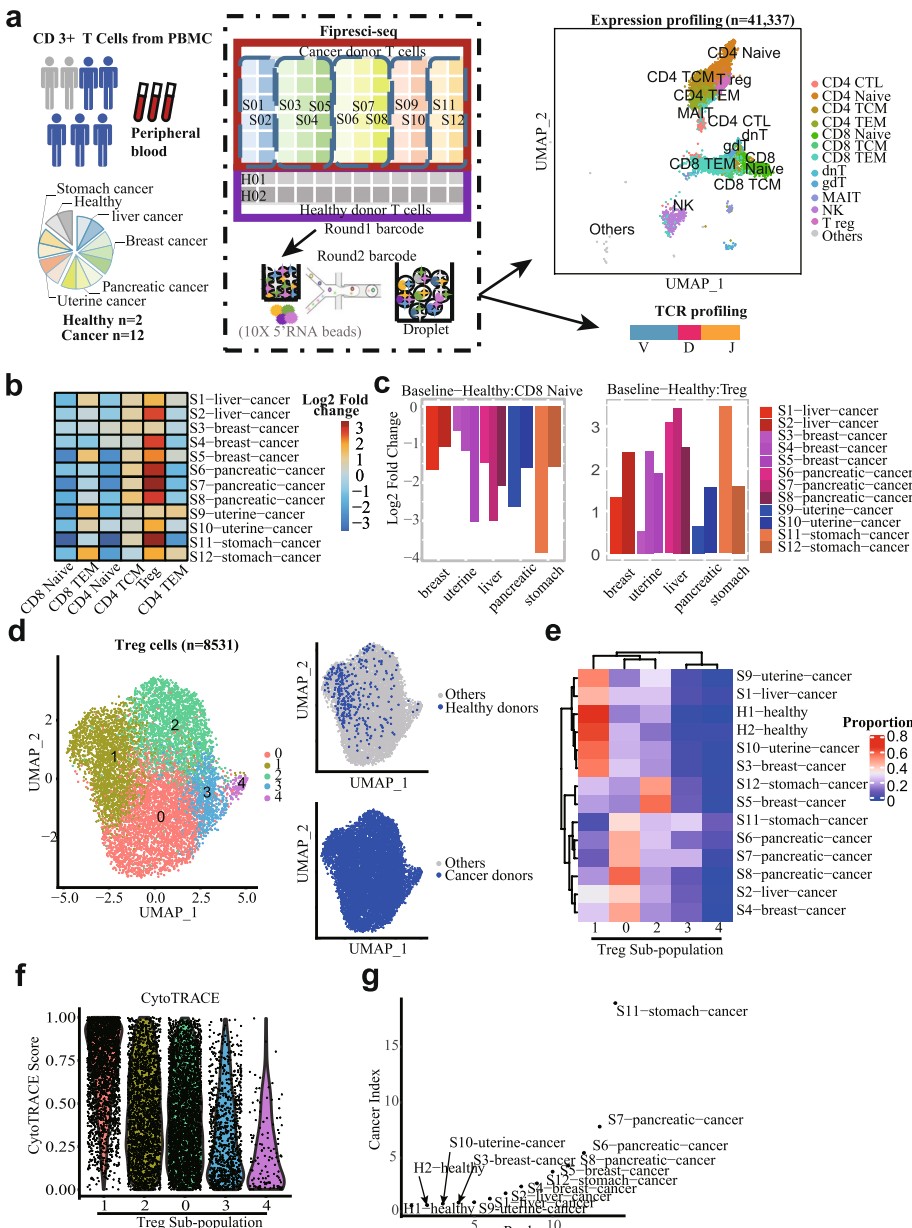

**Fig. 4** Single-cell transcriptome landscape of pan-cancer T cells FIPRESCI. **a** Flowchart depicting the overall experimental design of T cells (from peripheral blood of human donors (*n* = 14)) single-cell expression profiling and paired TCR profiling by FIPRESCI. UMAP plot (upper right) showing the major cell types detected. Total single cells (*n* = 41,377). **b** Heatmap showing the composition of the major T cell subtypes for different cancer patients and healthy donors. **c** Histogram showing the proportion change of the CD8 naïve (left) and T reg (right) in cancer patients compared to healthy donors. **d** Unsupervised clustering of Treg cells reveals 5 distinct Treg subpopulations. Left, UMAP embedding of 5 Treg subpopulation. Right top, UMAP plot colored by Treg cells from healthy donors. Right bottom, UMAP plot colored by Treg cells from cancer donors. **e** Heatmap showing five Treg subpopulations proportion over different donors. **f** Violin plot showing CytoTRACE Score distribution grouped by five Treg subpopulations. The order of subpopulations from left to right on the *X*-axis in descending order of mean CytoTRACE score within one subpopulation. **g** Dot plot showing the cancer index of each donor. Each dot represents one donor. *Y*-axis is the cancer index and *X*-axis is the rank according to the cancer index (the smaller the cancer index, the smaller the rank value)

Sample demultiplexing using two-round barcodes and quality-control filtering resulted in a final scRNA-seq dataset including 53,183 cells. It is known that distinct subpopulations of T cells, such as effector, regulatory, and γδ, are hard to be separated by scRNA-seq technologies alone. We, therefore, integrated our data with a well-anno-tated multimodal reference dataset for PBMC, in which subpopulations are jointly defined by mRNA and surface protein expression from CITE-seq with a "weighted-nearest neighbor" (WNN) analysis [27]. After applying the Seurat 4.0.5 [27] integra-tion procedure, 41,377 cells with high prediction scores were retained. Twelve types of T cells and NK cells appearing on the reference dataset were recovered in our data. These cell types include CD4 + T cells (4 clusters), CD8 + T cells (3 clusters), unconven-tional T cells (4 clusters), and NK cells (1 cluster) (Fig. 4a, right). It is encouraging that although the reference dataset includes all the cell types from PBMCs, 41,093 (99.3%) of our data are annotated as T cells, almost no B cells, dendritic cells, erythrocytes, and other monocytes, suggesting the high quality of annotation (Additional file 1: Fig. S7a-d). The expression of signature genes and known functional markers confirmed clusters of CD4 + (naïve, effector memory, central memory), conventional CD8 + (naïve, effector memory cells), and regulatory T cells (Treg) (Additional file 1: Fig. S8a-b). Interestingly, the surface proteins expression levels can be imputed from our data accurately when integrating with the PBMC CITE-seq reference dataset [27] (Additional file 1: Fig S9a). These results indicate that FIPRESCI scRNA-seq works well with primary human cells and can distinguish closely related cell types from complex human tissues or organs.

### Peripheral T cells exhibit a specific composition and transcriptome in cancer donors

We next compared the composition of the T cell subtypes detected in our data with the CITE-seq dataset from healthy donors [27]. The proportions of various cell types of our healthy donors were very similar to the reference PBMC atlas (Additional file 1: Fig. S9a-b). However, we observed that the frequencies of naïve T cells (including CD4 + and CD8 +) in cancer donors were much lower than that of healthy individuals, and the frequencies of Tregs were substantially higher in most cancer donors (Fig. 4b-c, and Additional file 1: Fig. S9a-c), especially in pancreatic cancer donors. Notably, Tregs were abundant in cancer donors' blood is consistent with the previous report based on flow cytometry [28, 29]. Moreover, Tregs frequency positively correlates with tumor metasta-sis and poor prognosis in human patients with pancreatic ductal adenocarcinoma (PDA) [30]. Our data confirmed that an abnormal proportion of T cells from peripheral blood may be a sign of cancer patients.

The FIPRESCI data enables identifying differential expression genes between healthy donors and cancer patients (Additional file 1: Fig. S10). Interestingly, some of the T cells' commonly expressed genes are specifically upregulated in certain cancers type compared with healthy donors. Such as CMSS1 and ZHX2 in uterine cancer, OSBPL8, FAM214A in breast cancer, TPT1, GPRIN3, and VAV3 in pancreatic cancer. Some genes are prefer-entially expressed in certain T cell subtypes but show a commonly dysregulated pattern in all cancers. For example, AOAH (a lipase acyloxyacyl hydrolase) and MAML2 (a tran-scriptional coactivator in the NOTCH-signaling pathway) are commonly downregulated in all cancers. On the contrary, CCL5 (a key T cell chemokine) is commonly upregulated in all cancers (Additional file 1: Fig. S10). These results indicated that FIPRESCI data

of T cell populations isolated from peripheral blood could be used to identify cancer-specific gene signatures for diagnosis and prognosis.

Next, we leverage the FIPRESCI single-cell data to explore the subpopulation difference of Treg cells between cancer and healthy donors. Unsupervised clustering of Treg cells from all donors reveals 5 clusters. Surprisingly, Treg cells from healthy donors are significantly enriched in cluster 1, while cells from cancer donors seem evenly distributed across all the clusters (Fig. 4d). Further breakdown of individual donors suggests that Cluster 1 is shared by healthy and cancer donors but mainly from 4 cancer donors, S9, S1, S10, and S3. Clusters 0,2,3,4 are almost exclusively contributed by cancer donors and collectively 2/3 of cancer donors have a distinct pattern compared to the healthy donor (Fig. 4e). There is strong heterogeneity across individual patients within the same cancer type except for pancreatic cancer. CytoTRACE [31] analysis suggests cluster 1 is a more naïve state while cluster 2,0,3,4 may represent terminally differentiated or effector states (Fig. 4f). We construct a "cancer index" based on the ratio of cluster1 and non-cluster1 cells, and the results suggest most cancer can be distinguished from healthy samples although a small proportion of cancer samples may lie in an intermediate zone (Fig. 4g). Tumor-infiltrating lymphocytes (TILs) are under intensive study [32], and T cells from cancer patients' peripheral blood are usually only used as a control to identify tumor environment-enriched T cell population. However, our analysis uncover previously unrecognized strong heterogeneity among cancer patients in terms of the extent of crosstalk between peripheral blood and tumor environment. In sum, our results highlight the importance of the classification of cancer patients based on the peripheral blood T cell population before using peripheral blood data as a control to study the TILs.

### FIPRESCI is compatible with single-cell immune repertoire profiling

We enriched the paired single-cell TCR VDJ sequences from FIPRESCI cDNA products of the above T cells by nested PCR according to the 10X Genomics manufacturer's protocol (Chromium Single Cell V(D)J Reagent Kits). After nested PCR enrichment, the first round of barcodes introduced by Tn5 tagmentation were eliminated. Therefore, in the VDJ-sequencing data analysis, droplets containing more than one cell (multiplets) were detected and removed based on two rounds of barcode information from corresponding scRNA sequencing data. As a result, TCR clonotypes were detected from 4936 cells across all 13 cell types after quality-control filtering (Fig. 5b, and Additional file 1: Fig. S12. a). We quantified V gene usage, and the top4 V gene segment usage observed no bias across each cell type (Fig. 5a). The V and J gene usage varied in cancer donors, while the use of the TRB V genes in two healthy donors was very similar (Additional file 1: Fig. S12a,b). We quantified the length distribution of the CDR3 sequences in each donor and found that the capture of TRA and TRB was unbiased across the donors (Additional file 1: Fig. S12c,d). Cells with the same CDR3 sequences for both the TCR α-chain and β-chain were defined as the same clonotype. We observed that clonotype composition in each sample varied by clone size. However, the proportion of the top 100 clone size has increased in most patients compared with healthy controls (Fig. 5e, Additional file 1: Fig. S12b).

Moreover, the number of unique clonotypes in each sample varied by cell type. However, compared with most cancer patients, the unique clonotypes of healthy

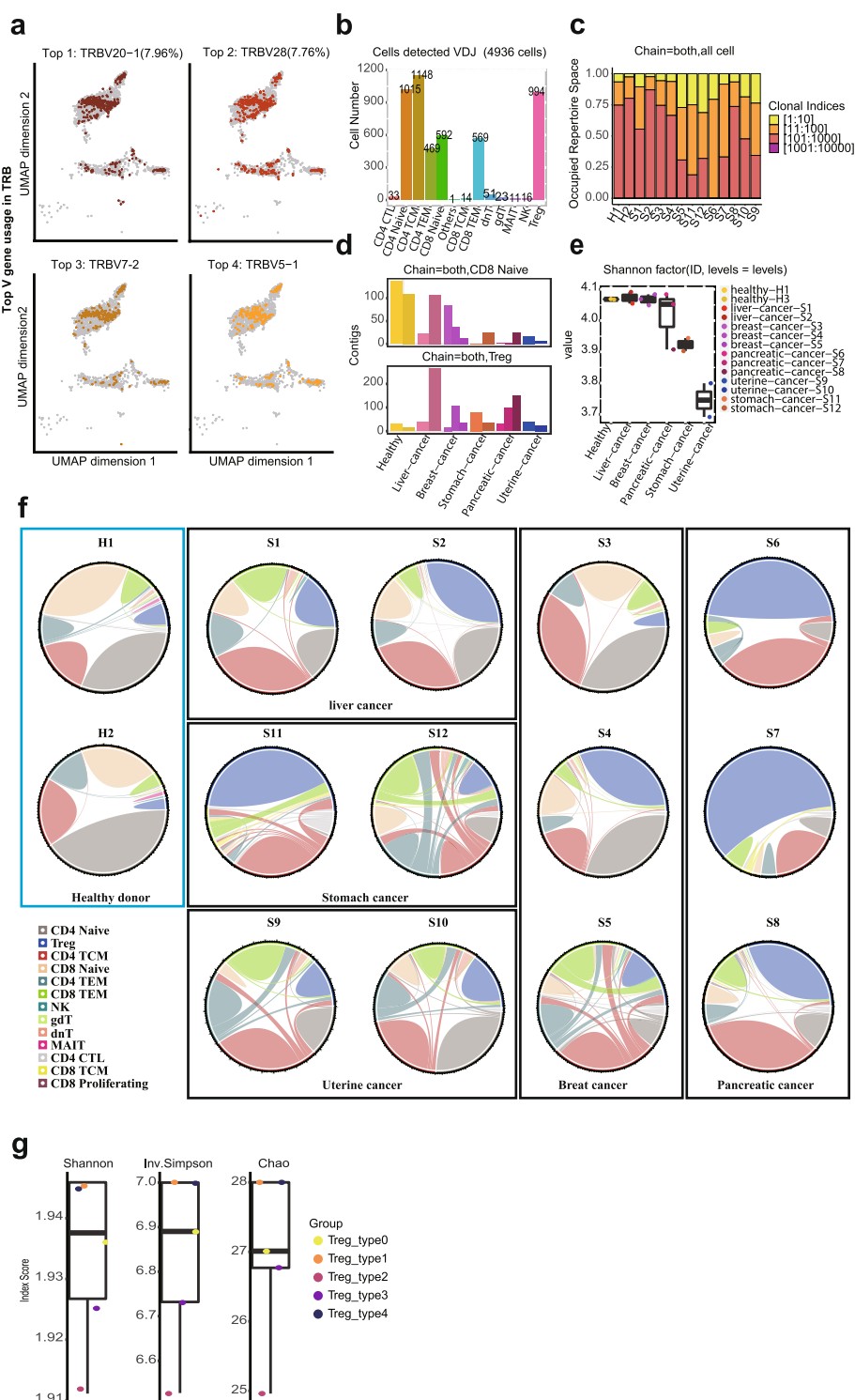

**Fig. 5** FIPRESCI is compatible with single-cell immune repertoire profiling. **a** UMAP shows the top 4 V genes usages in TRB. **b** Bar plots show the number of cells detected TCR VDJ by cell types (4,983). **c** Box plots showing the TCR clonal diversity. Diversity is calculated using Shannon metrics. **d** The histogram shows the distribution of the unique TCR clonotypes of the CD8 naïve (upper) and Treg (lower) across 14 donors. **e** Histogram showing the frequency and distribution of occupied TCR clonotypes across 14 donors. **f** Chord diagrams show the number of relative TCR clonotypes and the shared clonotypes across T cells in individual donors. **g** Box plot showing five Treg subpopulations clonotype diversity. Clonotype diversity is calculated as the Shannon index, inverse Simpson index, or Chao index within one Treg subpopulation

donors were abundant in naïve cells (including CD4 + and CD8 +), while fewer were in Treg cells (Fig. 5d, and Additional file 1: Fig. S12c). TCR clonal diversity was significantly lower in endometrial cancer and stomach cancer donors compared with healthy controls, while the diversity of breast cancer and liver cancer patients was comparable to that of healthy people, which may be due to these patients being in the early stages of cancer. (Diversity is calculated using two metrics: (1) Shannon, (2) inverse Simpson) (Fig. 5e, and Additional file 1: Fig. S12d).

Clonotypes were rarely shared across cancer patients (Additional file 1: Fig. S13a-c), and most clonotypes were singletons, representing a diverse repertoire of peripheral T cells. However, many clonotypes resided in multiple cell types in one patient, especially in stomach cancer patients, uterine cancer, and some breast cancer pancreatic patients (Additional file 1: Fig. S13c), indicating the clonal expansion of T cells in these patients. Furthermore, the proportions of different types of cells detected with TCR clonotypes varied greatly among donors (Fig. 5f). Compared with healthy donors, most patients showed abundant Treg which is consistent with previous transcriptome-based composition analysis. What is more, we found that there are two TCR sequences from patient S2 and one TCR sequence from patient S5 are clustered together with NeoTCRs recently reported (Additional file 1: Fig. S13d), raising the possibility of identification of cancer-reactive T cells in peripheral blood by FIPRESCI with deeper sampling. These results suggested that FIPRESCI enables single-cell VDJ analysis and provides a critical additional layer of information of clonotype and antigen specificity to study cancer-related T cells.

## Discussion

In summary, we presented FIPRESCI, a simple, efficient, and massive-scale single-cell transcriptomics approach that is compatible with both fresh or fixed cells and nuclei. In addition, FIPRESCI specifically captures the 5′-end of the transcript, which contains rich information for studying gene regulation as it enables the identification of active TSSs and detects distal transcribed cis-regulatory elements. Only ~ 2% of FIPRESCI data are contributed by eRNA reads, and there may be many eRNAs that are too short to be detected. However, thousands of pairs of eRNA and protein-coding genes can be observed co-expressed in the same single cell, which may provide a unique opportunity to explore gene regulation. Further sophisticated deep learning prediction or imputation models based on multi-omics reference datasets may leverage the scalability of FIPRESCI and extend the utility of data to other modalities, such as chromatin accessibility or protein expression. Moreover, the single-cell RNA-5′end-sequencing method provides a widely used strategy for profiling full-length immune receptor repertoire information and gene expression from the same cell. So far, there is only a limited number of scRNA-seq protocols designed to capture the 5′-end of the transcript. Most of them are low throughput or costly. We demonstrated FIPRESCI can increase throughput by at least tenfold over the standard 10X Genomics 5′ scRNA-seq procedure. We applied FIPRESCI to investigate gene expression and TCR repertoires at single-cell resolution for T cells from PBMC of 14 donors across 5 cancer types and healthy individuals. Distinct composition, transcriptomic features, and TCR clonotype of T cell subtypes across cancer patients were identified from our data. A single experiment of FIPRESCI

can be completed by a single person in 12 h. Thus, it will accelerate the large-scale study of the profile of the transcriptomes, regulomes, and immune receptor repertoires from millions of single cells or thousands of samples.

Similar to the previous report, we found methanol fixation does not affect the cell type clustering, but certain types of biological analyses that may be influenced by the GC content or transcript length need to use caution when interpreting the results from FIPRESCI data with methanol fixation [33]. Our approach adds a seamless whole transcriptome single-molecule preindexing step to standard droplet microfluidic 5′ sc(n)-RNA-seq workflow, thus is complementary to other efforts which try to enhance the throughput, flexibility, or efficiency. FIPRESCI is a single-cell combinatorial indexing strategy using droplet microfluidic. It is conceptually distinct from sample multiplexing approaches which can only increase the cell throughput moderately as the data from doublets are not resolved but discarded. FIPRESCI can make the data from doublets useful thus increasing the cell throughput exponentially. Although single-cell combinatorial indexing has built-in support for massive sample multiplexing by using the first round barcode to record the sample identity, a recent study shows that it can be beneficial to incorporate single-cell combinatorial indexing with a sample labeling strategy(nuclear hashing) to achieve high efficiency [34]. In addition, the optimization of the reagent chemistry (e.g., the 5′ Library & Gel Bead Kit v1 upgrade to v2) and increasing the number of the droplets (e.g., Chromium Controller upgrade to Chromium X) can straightforwardly be incorporated into the current FIPRESCI workflow without further configuration. FIPRESCI is expected to support multi-omics assays, such as simultaneous expression profiling of the transcriptome and surface proteins from the same cell [15], antigen specificity [35], CRISPR screening, and spatial transcriptomics [36, 37]. In principle, it is feasible to adapt FIPRESCI to other microfluidic or microwell platforms which support TSO barcoding based 5′-end sc(n)-RNA-seq.

## Conclusions

In conclusion, due to the low cost and ultra-high throughput of FIPRESCI, it is suitable for multi-sample and large-scale single-cell transcriptomics profiling, especially when capturing the 5′-end of transcript is important. FIPRESCI enables us to acquire multiple layers of additional information from a single scRNA-seq experiment with high efficiency. We expect that FIPRESCI can be applied immediately to a wide range of fields to provide important insights, such as large-scale cell atlas studies, organ development studies across time and space, single-cell immune receptor repertoire studies for large-scale cohorts of diseases (e.g., cancer, autoimmune, severe pandemics such as Coronavirus disease 2019), high-throughput CRISPR, and/or drug screens.

## Methods

### Antibodies and staining solution

Antibodies and reagents used were APC anti-human CD3 Antibody (1:20, BioLegend, 317,318) and 7-AAD Staining Solution (1:50, Abcam, ab228563).

## Cell culture

All established cell lines were purchased from the National Collection of Authenticated Cell Cultures (Shanghai, China). Cells were cultured at 37 ℃ in an atmosphere of 5% (v:v) carbon dioxide in DMEM (NIH-3T3, HeLa, and HEK293T) or RPMI (K562, Jurkat) supplemented with 10% fetal bovine serum.

## Indexed Tn5 transposome assembly

The assembly of the i7-only Tn5 transposome was performed as per the manufacturer's instructions for TruePrep Tagment Enzyme (Vazyme, S601-01) reagent. TN5_A_ME and TN5_R2_index (sequences are provided in Supplementary Table 1) were synthesized and purified by high-performance liquid chromatography (Sangon Biotech (Shanghai)) and dissolved into Annealing Buffer (Vazyme, S601-01) at a final stock concentration of 10 μM. For the annealing reaction, oligonucleotides were mixed at a 1:1 molar ratio at 10 μM and mix them thoroughly and anneal samples using the following thermocycling parameters: 75 ℃ for 15 min; 60 ℃ for 10 min; 50 ℃ for 10 min; 40 ℃ for 10 min; 25 ℃ for 30 min. After this step oligonucleotide cassette can be aliquoted and frozen at − 20 ℃ for future transposome assemblies. To assemble the Tn5 transposase, we mixed 7 μl of oligonucleotide cassette from the previous step with 4 μl of TruePrep Tagment Enzyme (Vazyme, S601-01), and 39 μl coupling buffer (Vazyme, S601-01), mixed well and then incubated for 1 h at 30 ℃ in a thermocycler. The resulting 50 μl of assembled transposome can be stored at − 20 ℃ for at least 1 month.

## Assays of Tn5 in vitro tagmentation on RNA/DNA hybrids

### Messenger RNA preparation

Total RNA was extracted from HEK293T cells with TRIzol (Invitrogen, 15,596,026), according to the manufacturer's instructions. The resulting total RNA was treated with DNase I (NEB, M0303) to avoid genomic DNA contamination. Phenol/chloroform extraction and ethanol precipitation were then performed to purify and concentrate total RNA. For mRNA isolation, mRNA Capture Beads (Vazyme, N401-01) were used according to the manufacturer's instructions. Cre mRNA was prepared by in vitro transcription of synthetic Cre DNA with HiScribe T7 ARCA mRNA Kit (NEB, E2060S) according to the manufacturer's instructions, then treated with DNase I (NEB) to digest DNA template.

### mRNA/DNA hybrids preparation

Hela mRNA and Cre mRNA were reverse transcribed into RNA/DNA hybrids separately by Maxima H Minus Reverse Transcriptase (Invitrogen, EP0752), according to the manufacturer's protocol. mRNA/DNA hybrids products were purified with 0.6X Agencourt AMPure XP SPRI beads (Beckman, A63881), according to the manufacturer's protocol.

### Tn5 in vitro tagmentation on RNA/DNA hybrids

For testing whether or not the Tn5 transposase also stayed bound to its RNA/cDNA substrate after transposition, we added a transposition mix (4 μl 5×Reaction buffer from TruePrep ® Tagment EnzymeTn5 transposase kit (Vazyme, S601-01), $H_2O$, 2 μl 10 μM assembled transposome) to 50 ng Hela and Cre RNA/DNA hybrid products (in a final volume of 20 μl) respectively and mixed gently by pipetting. Each RNA/DNA hybrid

performed 3 tubes of the test. The transposition tubes were incubated at 55 °C for 10 min in a thermocycler with a heated lid. For 3 tubes:

(1) One of the transposition products (10 μl) was added with 10 μl of 40 mM EDTA (pH8.0) and then stored on ice;

(2) One was added with 2 μl of 1% SDS and incubated at RT for 10 min and then stored on ice;

(3) The third one was added with 2 μl of 1% SDS and purified by using 1.2X Agencourt AMPure XP SPRI beads to remove Tn5 and excess free adaptors, and eluted in 15 μl nuclease-free water. Then added 25 μl NEBNext High-Fidelity $2 \times$ PCR Master Mix (NEB, M0541S), 5 μl 10 μM S-P7 (CAAGCAGAAGACGGCATA CGAGATGTGACTGGAGTTCAGACGTGTGCTCTTCCGATC-s-T), and 5 μl nuclease-free water to the transposed RNA/DNA hybrids. The linear amplification PCR was performed in a thermocycler as follows:72 °C for 3 min, 98 °C for 45 s, 11 cycles [98 °C for 20 s, 67 °C for 30 s, 72°for 1 min], 72 °C for 1 min, storage at 4 °C. Then the PCR product was cleaned with $0.8 \times$ AMPure XP SPRI beads (Beckman Coulter, A63881), eluting in 20 μl of EB buffer.

### PAGE analysis

The quality of the complexes was assessed on an 8% TBE gel (Invitrogen, EC62155BOX) according to the manufacturer's instructions.

### FIPRESCI methods

**Note:**

(1) Sequences of oligos or primers used below are all provided in Supplementary Table 1.

(2) The round1 barcode information involved in each FIPRESCI experiment mentioned in the manuscript is provided in Supplementary Table 2.

### Single-cell suspension preparation

The single-cell suspension was further processed into nuclei or permeabilized cells.

### Nuclei isolation

Cells (30,000–3,000,000) were washed with 3–10 ml of ice-cold $1 \times$ PBS (hyclone, SH30256.01) and centrifuged at $300 \times g$ for 5 min at 4 °C. The cell pellets were resuspended in 100–1000 μL chilled lysis buffer (10 mM Tris–HCl, pH 7.4 (Sigma, T2194), 10 mM NaCl (Sigma, 59222C), 3 mM $MgCl_2$ (Sigma, M1028), 0.1% NP40 (Sigma, 74,385), 0.1% Tween-20 (Thermo, 28,320), 0.01% digitonin (Thermo, BN2006), 1 U/μl Sigma Protector RNase inhibitor (Sigma, 3,335,399,001) with 1% BSA (Sigma, A1933)), and mixed 10 times with a pipette. After incubation for 3–5 min on ice, 1–10 ml chilled Wash Buffer was added (10 mM Tris–HCl, pH 7.4, 10 mM NaCl, 3 mM $MgCl_2$, 0.1% Tween-20, 1% BSA) to the lysed cells. Cell suspension was mixed

5 times with pipette, then centrifuged at 300 *g* for 5 min at 4 °C and resuspended in 5–500 µl of ice-cold PBS-BSA- RNase inhibitor (1 × PBS supplemented with 1% w/v BSA (Sigma, A1933) and 1 U/µl Sigma Protector RNase inhibitor (Sigma, 3,335,399,001)), and concentration was determined by using a LUNA™ Automated Cell Counter.

### Cell permeabilization

Cells (30,000–3,000,000) were washed with 3–5 ml of ice-cold 1 × PBS (Hyclone # SH30256.01) and centrifuged at 300 × *g* for 5 min at 4 °C. The cell pellets were re-suspended in 100 µl 1 × PBS and fixed in 900 µl of ice-cold methanol (Fisher Scientific #M/4000/17) at − 20 °C for 10 min. After two additional washes (centrifugation: 300 × *g*, 5 min, 4 °C) with 1 ml of ice-cold PBS-BSA- RNase inhibitor, the permeabilized cells were filtered through a cell strainer (40 or 70 µM depending on the cell size) then re-suspended in 5–500 µl of ice-cold PBS-BSA-SUPERase and counted by using a LUNA Automated Cell Counter (Luna-FL).

### Reverse transcription

One hundred thousand cells or nuclei (7 µl) were added with 3 µl 25 µM RT primers (containing RT PolyT Primer (NEB, S1327S), or RT random Primer (Thermo Scientific, SO142), or their mix)), incubated for 5 min at 55 °C to resolve RNA secondary structures, then placed immediately on ice to prevent their re-formation. Then a mix of 40 µl RT reaction buffer, containing 10 µl 5 × Reverse Transcription Buffer, 2.5 µl of 100 mM DTT (Thermo Fisher Scientific, P2325), 2.5 µl of 10 mM dNTPs (Vazyme, P031-01), 2.5 µl of RNaseOUT RNase inhibitor (40 U/ml, Thermo Fisher Scientific, 10,777,019), 3.5 µl of Maxima H Minus Reverse Transcriptase (200 U/ml, Thermo Fisher Scientific, EP0753), and 21.5 µl nuclease-free water were added. The reverse transcription was incubated as follows (with a heated lid set to 60 °C): 50 °C for 10 min, 3 cycles of [8 °C for 12 s, 15 °C for 45 s, 20 °C for 45 s, 30 °C for 30 s, 42 °C for 2 min, 50 °C for 3 min], 50 °C for 5 min, stored at 4 °C.

### i7 only transposome tagmentation

Cells or nuclei were then distributed into one or multiple 96-plates (each well was already added with18µl transposition mix, containing 1 × Tagmentation buffer (the detailed components of tagmentation buffers for individual experiments are provided in Supplementary Table 3), and 1 µM indexed transposome). For each well, 2000–4000 cells or nuclei (1–2 µL) were added and pipetted to mix thoroughly, then incubated in a thermomixer at 1000 rpm for 30 min at 37℃. Tagmentation was stopped by adding 5 µl of 0.5 M EDTA to quench TN5, and the sample was held on ice for 10 min.

### Sample pooling

Then the cells or nuclei were pooled together, and 240 µl 10% BSA was added (for a final 1% BSA) to prevent cells or nuclei from clumping during the following centrifugation step. Wells were washed with ice-cold 1 × PBS containing 1% BSA, which was transferred to the same tube for maximum recovery. Samples were centrifuged at 600 *g* for

5 min, supernatant was removed, and pellets were washed twice with 0.5 ml of ice-cold PBS-BSA-RNase inhibitor, then resuspended in 20 μl of PBS-BSA-RNase inhibitor.

### GEM generation and cDNA template switch

Cells or nuclei pools were then processed on a single 10 × microfluidics lane. Sequencing libraries were prepared using a custom protocol based on the 10X Genomics Single Cell 5′ RNA Reagent Kit (10X Genomics, Pleasanton, California) workflows. Briefly, the 10 × workflow was followed up until single-cell Gel Bead-In-Emulsions (GEM) generation. The aimed target cell recovery for each library was based on customer requirements (according to our test, the targeted nuclei recovery rate was above 58%). In brief, cellular suspensions were loaded on the sample chip in the Chromium Controller instrument (10X Genomics) according to the manufacturer's instruction (for Chromium Next GEM Single Cell V(D)J Reagent Kits v1.1 refer to User Guide: CG000207 Rev E, or Chromium Next GEM Single Cell 5′ v2 refer to User Guide: CG000331 Rev A) to generate GEMs. After the Chromium controller completed running, 100 μl GEMs were transferred into the tube strip on ice with the pipette tips against the sidewalls of the tubes, then incubated in a thermomixer for 30 min at 25 °C, 90 min at 42 °C, and 10 min at 53 °C.

### GEM cleanup

The emulsion was broken and purified according to the manufacturer's instructions (10X Genomics). Briefly, 125 μl Recovery Agent (10X Genomics catalog no. 220016) was added to each sample (post GEM-RT incubation) at room temperature for 2 min. Then 125 μl of the pink oil phase was removed by pipetting. The remaining sample was mixed with 200 μl of Dynabead Cleanup Master Mix (per reaction: 182 μl of Cleanup Buffer (10X Genomics, 2,000,088), 8 μl of Dynabeads MyOne Silane (10X Genomics, 2,000,048), 5 μl of Reducing Agent B (10X Genomics, 2,000,087), and 5 μl of nuclease-free water). After 10 min of incubation at room temperature, samples were washed twice with 300 μl of freshly prepared 80% ethanol (Merck, 603–002-00–5) and eluted in 35.5 μl of EB Buffer (Qiagen, 19,086) containing 0.1% Tween (Sigma, P7949-500ML) and 1% v/v Reducing Agent B. Bead clumps were sheared with a 10-μl pipette or needle. Thirty-five μl of the sample was transferred to a new tube.

### cDNA enrichment

Thirty-five microliters of the above product were mixed with 65 μl PCR reaction mix, containing 50 μl of NEBNext High-Fidelity 2 × PCR Master Mix (NEB, M0541S), 0.5 μl of 100 μM S5R-P5-Bio primer (5′Biotin-AATGATACGGCGACCACCGAGATCTAC ACTCTTTCCCTACACGACGCTC-3′), and 14.5 μl of nuclease-free water. 5′-end cDNA was amplified in a thermocycler as follows: 72 °C for 3 min, 98 °C for 45 s, 13–16 cycles (based on input cell number of [98 °C for 20 s, 67 °C for 30 s, 72 °C for 1 min], 72 °C for 1 min, storage at 4 °C). Then the cDNA enrichment reaction was cleaned with 0.8 × AMPure XP SPRI beads (Beckman Coulter, A63881), eluting in 40 μl of EB buffer. The product was then purified with Dynabeads MyOne Streptavidin C1 (Invitrogen, 65,001). In brief, 10 μl of Dynabeads MyOne Streptavidin C1 was washed twice with 1 × B&W buffer (5 mM.

Tris pH 8.0, 1 M NaCl, 0.5 mM EDTA). After washing, the beads were resuspended in 40 µl of 2 × B&W buffer (10 mM Tris pH 8.0, 2 M NaCl, 1 mM EDTA) and mixed with the sample. The mixture was rotated on an end-to-end rotator at 10 rpm for 60 min at room temperature. The lysate was put on a magnetic stand to separate the supernatant and beads, then washed beads with 100 µl 1 × B&W buffer, and 100 µl elution buffer (Qiagen, 19,086) again. Finally, beads were resuspended in 20 µl nuclease-free water.

### Library preparation and sequencing
Twenty microliters of enriched cDNA products (with C1 Beads) were then mixed with 80 µl library reaction mix, containing 50 µl of 2X KAPA HiFi HotStart Ready Mix (Kapa, KK2602), 0.5 µl of 100 µM S5R-P5 primer (5′-AATGATACGGCGACCACCGAGATC TACACTCTTTCCCTACACGACGCTC-3′), 5 µl of 10 µM S-P7-index primer (5′-CAAGCAGAAGACGGCATACGAGAT[NNNNNN]GTGACTGGAGTTCAGACG TGTGCTCTTCCGATC-s-T-3′), and 24.5 µl of nuclease-free water. The reaction was amplified in a thermocycler as follows: 98 °C for 45 s, 16 cycles of [98 °C for 20 s, 54 °C for 30 s, 72 °C for 20 s], 72 °C for 1 min, storage at 4 °C. Libraries were purified with 0.75 × AMPure XP SPRI beads and eluted with 24.5 µl nuclease-free water. The final libraries were quantified using a Bioanalyzer High-Sensitivity DNA chip (Agilent, 5067–4626) and Qubit HS assay (Thermo Fisher Scientific, Q32854) and then sequenced in NovaSeq 6000 (Illumina, San Diego, CA) or MGISeq-T7 (MGI Tech Co., Ltd., China) with a 150-bp paired-end read length, targeting a depth of 10,000–50,000 reads per cell.

### Species-mixing controls
Human (Jurkat) and mouse (NIH-3T3) cell lines were processed using the FIPRESCI protocol as described above. Briefly, methanol-fixed whole cells were prepared separately for each of the cell lines. Prepared Jurkat and NIH/3T3 cells were mixed at a 1:1 ratio, and reverse transcription was performed. After the RT reaction, cells were tagmented Tn5 on a 96-well plate loaded with indexed oligonucleotides (containing unique sets of round1 indices) (Supplementary Table 2). For 10X Genomics Chromium processing, 15,300 cells per microfluidic channel were loaded to generate single-cell Gel Bead-In-Emulsions (GEM) using Chromium Next GEM Single Cell V(D)J Reagent Kits v1.1 according to User Guide: CG000207 Rev E.

### scRNA-seq and snRNA-seq experiment for FIPRESCI
For the proof-of-concept scRNA-seq and snRNA-seq experiments, HEK293 cells, K562 cells, and Hela cells were prepared nuclei and methanol-fixed, permeabilized whole cells separately for each of the cell lines. scRNA-seq and the snRNA-seq experiment were processed separately using the FIPRESCI protocol as described above. Briefly, methanol-fixed whole cells or nuclei were subjected to reverse transcription separately for each of the cell lines. After the RT reaction, different cell lines were loaded into TN5 wells (containing unique sets of round1 indices) as Supplementary Table 2 described. For 10X Genomics Chromium processing, 100,000 cells or nuclei were loaded per microfluidic channel to generate single-cell Gel Bead-In-Emulsions (GEM) using Chromium Next GEM Single Cell V(D)J Reagent Kits v1.1 according to User Guide: CG000207 Rev E.

**Alternative buffers for tagmentation activity of FIPRESCI**

For testing the tagmentation activity of Tn5 on RNA/DNA hybrids within permeabilized cells or nuclei, we performed a set of experiments with different 16 reaction buffers and variations according to the previously published studies (as described in the manuscript). Hela and HEK293 methanol-fixed permeabilized cells were used separately to test the performance of each tagmentation buffer (tagmentation buffers used in the manuscript are provided in Supplementary Table 3). Briefly, methanol-fixed Hela or HEK293 cells were reverse transcribed separately. After the RT reaction, different cell lines were loaded into TN5 wells containing different tagmentation buffers and unique sets of round1 indices as Supplementary Table 2 describes. For 10X Genomics Chromium processing, 15,300 HEK293 cells and 15,300 Hela cells were loaded per microfluidic channel to generate single-cell Gel Bead-In-Emulsions (GEM) using Chromium Next GEM Single Cell V(D)J Reagent Kits v1.1 according to User Guide: CG000207 Rev E.

**E10.5 mouse embryo nuclei with FIPRESCI readout**

For the collection of E10.5 mouse embryos, 8–10-week-old female C57BL/6 mice (purchased from Beijing Vital River Laboratory Animal Technology, Beijing, China) were mated to 10–12-week-old male PWK/PhJ mice (purchased from Jackson Laboratory) naturally, and the first day that the vaginal plug was observed was considered as E0.5. We collected 2 embryos. Embryos were washed twice with DPBS and were cut into small pieces. Then tissues were digested with 1 mg/ml type II collagenase (Gibco, 17,101,015) and 1 mg/ml type IV collagenase (Gibco, 17,104,019) at 37 °C for 40 min.

Dissociated cells were centrifuged at 300 *g* for 5 min at 4 °C, then resuspended in 1 mL of cold DPBS with 0.1% BSA. After passing through a 40-μm strainer (Biologix), cells were washed twice, centrifuged at 300 *g* for 5 min at 4 °C, resuspended in cold DPBS with 0.1% BSA at a density of $1 \times 10^5$ cells/ml, and stored on ice before scRNA-Seq and nuclei isolation.

To isolate nuclei, the cell pellets were resuspended in 200 μL chilled lysis buffer (10 mM Tris–HCl, ph 7.4, 10 mM NaCl, 3 mM $MgCl_2$, 0.1% NP40, 0.1% Tween-20, 1 mM DTT, 1 U/μl Protector RNase inhibitor (Sigma, 3,335,402,001), 0.01% digitonin, and supplemented with 1% BSA), and pipette mixed $10 \times$. After incubation for 5 min on ice, 1 ml chilled Wash Buffer was added (10 mM Tris–HCl, ph 7.4, 10 mM NaCl, 3 mM $MgCl_2$, 0.1% Tween-20, 1 mM DTT, 1 U/μl Protector RNase inhibitor (Sigma, 3,335,402,001), 1% BSA) to the lysed cells. After pipette mixed $5 \times$, nuclei were collected by centrifugation (500 *g*, 5 min, 4 °C) and fixed in 1 ml of ice-cold $1 \times$ PBS containing 2% formaldehyde (Thermo Fisher Scientific, 28,908) for 10 min on ice. Fixed nuclei were collected (500 *g*, 5 min, 4 °C), and the pellet was resuspended in 1.5 ml of ice-cold PBS-BSA-SUPERase ($1 \times$ PBS supplemented with 1% w/v BSA (Sigma, A8806-5) and 1% v/v SUPERase-In RNase Inhibitor (Thermo Fisher Scientific, AM2696)) and nuclei were collected (500 *g*, 5 min, 4 °C). After one more wash with 1.5 ml of ice-cold PBS-BSA-SUPERase, fixed nuclei were resuspended in chilled Diluted Nuclei Buffer (10X Genomics, PN-2000153) based on the number of cells used for isolation and assuming $\sim 50\%$ nuclei loss during cell lysis, and transferred to a 1.5-ml tube. If cell debris and large clumps are observed, they are pass through a cell strainer.

Fixed nuclei were then processed using the FIPRESCI protocol as described above. Briefly, 400,000 fixed nuclei were used for the reverse transcription reaction. After the RT reaction, nuclei were gently pipetted and then loaded into TN5 wells (containing unique sets of round1 indices). After tagmentation, stopping the reaction, pooling the sample, and washing, about 200,000 nuclei were remaining. For 10X Genomics Chromium processing, all the remaining nuclei were loaded to 10X Genomics microfluidic channel for generating single-cell Gel Bead-In-Emulsions (GEM) using Chromium Next GEM Single Cell 5′ v2 according to User Guide: CG000331 Rev A.

### Preparation of human T cells for FIPRESCI

Whole blood was obtained from fourteen donors, including two healthy people and twelve patients who were pathologically diagnosed with cancer (including 5 cancer types: endometrial cancer, breast cancer, liver cancer, pancreatic donors, and stomach cancer), who were enrolled in this study. These samples were collected from the Chinese People's Liberation Army General Hospital. The available clinical characteristics of twelve patients are summarized in Supplementary Table 4.

The fresh peripheral blood was collected in EDTA anticoagulant tubes and processed immediately. For each donor, Peripheral blood mononuclear cells (PBMCs) were isolated according to the manufacturer's instructions for Ficoll-Paque PLUS (GE Healthcare, 17–1440-02). Briefly, 2 ml of fresh peripheral blood were combined, diluted in 2 ml $1 \times$ PBS, and carefully added to a 15-ml tube containing 2 ml of Ficoll. The tubes were centrifuged for 20 min at 1000 $g$ (minimum acceleration and deceleration). The interphase was carefully collected and washed with PBS and subsequently centrifuged for 10 min at 250 $g$ at room temperature. Isolated PBMCs were gently suspended with cryopreservation medium CELLBANKER2 (AMSBIO, 11,891), and the cell suspension was dispensed in 100 μl aliquots to cryopreservation vials. The vials were stored directly at $-80℃$ before further processing.

After isolating PBMCs from all samples, we used FACS to sort T cells for sequencing. Single-cell suspensions were subjected to antibody staining with anti-CD3 and 7-AAD for FACS sorting performed on a BD Aria SORP instrument. FACS gates were drawn to include only live single cells based on 7-AAD. Further gates were drawn to arrive at CD3+. Based on FACS analysis, live single T cells were sorted into 1.5-mL tubes (Eppendorf). The sorted cells were placed on ice until all samples were prepared.

Human T cells were then processed using the FIPRESCI protocol as described above. Briefly, methanol-fixed whole cells for reverse transcription were prepared separately for each of the donors (added 15,000 cells per donor). After the RT reaction, different donors' cells were loaded into TN5 wells (containing unique sets of round1 indices) as Supplementary Table 2 describes. For 10X Genomics Chromium processing, 80,000 cells per microfluidic channel were loaded to generate single-cell Gel Bead-In-Emulsions (GEM) using Chromium Next GEM Single Cell 5′ v2 according to User Guide: CG000331 Rev A.

### Single-cell VDJ enrichment from T cell cDNA product of FIPRESCI

VDJ libraries were enriched from the pan-cancer T cell cDNA product of FIPRESCI by using two consecutive nested PCRs and VDJ library preparation according to the manufacturer's protocol (Chromium Single Cell V(D)J Reagent Kits, 10X Genomics). The final libraries were quantified using a Bioanalyzer High-Sensitivity DNA chip (Agilent, 5067–4626) and Qubit HS assay (Thermo Fisher Scientific, Q32854) and then sequenced in NovaSeq 6000 (Illumina, San Diego, CA) with a 150-bp paired-end read length, targeting a depth of 5000 reads per cell.

### Bulk ATAC-seq

10,000 Hela nuclei were used to generate bulk ATAC-seq library with TruePrep DNA Library Prep Kit V2 for Illumina kit (vazyme, TD501) according to the manufacturer's protocol.

### Processing of FIPRESCI data

FASTQ files generated from FIPRESCI were demultiplexed according to round1 barcodes, any mismatch was not allowed in this step. After demultiplexing, FASTQ files with the same round1 barcode can be regarded as the output of standard 10X genomics single-cell 5-end RNA-seq experiments. Thus, reads containing the same round1 barcode were used as inputs for 10X Genomics software Cell Ranger (version 6.0.2, parameters: –chemistry = "fiveprime", –include-introns). To generate the final gene expression matrix, we merged Cell Ranger output filtered_feature_bc_matrices from different round1 barcodes by Seurat (version 4.0.5) [27] "merge" function. Round1 barcodes were added to their corresponding round2 (10X genomics) barcodes through this step, forming cell barcodes. Therefore, cell barcodes and the final gene expression matrix were obtained. Finally, we saved that gene expression matrix as a final matrix in.rds format for downstream analysis.

### Species mixture FIPRESCI data analysis

We used Seurat (version 4.0.5) for the Species Mixture FIPRESCI data analysis. Firstly, cells whose UMI < 1000 were filtered. To normalize gene expression levels, gene expression values for each cell were divided by total counts for that cell and multiplied by 10,000 then log transformed. PCA (principal component analysis) was performed after scaled and centered data. Dimensionality reduction method UMAP (Uniform Manifold Approximation and Projection) was performed by the Seurat function "RunUMAP" with the top 20 PCs and other default parameters. To distinguish mouse cells or human cells from mouse and human mixture data, we defined a cell as a mouse cell if its mm10 UMI count percentage > 90%, and a cell as a human cell if its mm10 UMI count percentage < 10%.

### Nuclei and permeabilized prepared three cell line FIPRESCI data analysis

We analyzed nuclei or permeabilized prepared three cell line FIPRESCI data independently by Seurat (version 4.0.5). Similarly to species mixture FIPRESCI data analysis, after normalization and PCA, the top 10 PCs were used as input for the dimensional

reduction method tSNE (t-distributed Stochastic Neighbor Embedding). For comparative analysis of the two kinds of cell preparation, firstly we selected the top 3000 highly variable genes from each cell preparation condition. Secondly, we used intersected highly variable genes as common features. A gene's expression profile within one cell line of one cell preparation condition was calculated as the mean value of UMI counts. Finally, Spearman correlation was used to evaluate similarity across different cells prepared the three cell line FIPRESCI profile. To identify differentially expressed genes, we performed a Wilcox test and calculated the average log2 fold change value between two groups of cells using the Seurat function "FindMarkers".

### Sensitivity analysis across different RT primer and cell preparation conditions

For sensitivity analysis across 6 different RT primer and cell preparation conditions, we selected 6 different round1 barcodes (each round1 barcode represented one RT primer and cell preparation condition) and their corresponding read files. For each condition, we randomly downsampled read pairs and ensured that mean read pairs per cell are 10 k, 20 k, 30 k, 40 k, and 50 k by seqtk (parameters: -s100) (https://github.com/lh3/seqtk). Then, we mapped each downsampled read pair to the genome independently and calculated the average detected gene numbers per cell.

### TSS enrichment and enhancer enrichment analysis for RT primer condition FIPRESCI data

We used deeptools (version 2.29.0) [38] and bedtools (version 3.3.0) [39] for read enrichment analysis. Firstly, bam files were removed PCR duplication by Picard (version 2.21.6) (https://broadinstitute.github.io/picard/). Then, we used the deeptools command "bamCoverage" (parameters: −normalizeUsing RPKM, −smoothLength = 200) to generate a read coverage profile across the genome. For TSS enrichment analysis, we used the deeptools command "computeMatrix reference-point" (parameters:−referencePoint TSS, -a 2000 and -b 2000) to calculate read coverage around TSS. For enhancer enrichment analysis, we used macs2 (version 2.2.7.1, parameters: −shift -100, −extsize 200, −nomode and -q 0.05) [40] to call ATAC peaks from bulk Hela ATAC-seq data. Then, enhancers were defined as ATAC peaks whose distance to all TSS is greater than 2000 bp. We used deeptools "computeMatrix reference-point" command (parameters: −referencePoint center, -a 2000 and -b 2000) to calculate read coverage around the enhancer center. Finally, we used the deeptools command "plotHeatmap" and "plotProfile" to visualize the results.

### Identification of eRNA transcription start regions

We identified eRNA by FIPERSCI data and corresponding ATAC-seq. We defined distal ATAC-seq peaks as those whose distance to any TSS is greater than 2000 bp. Then we calculate the RPKM (reads per kilobase of transcript per million reads mapped) value for each distal ATAC-seq peak using FIPERSCI sc(n)RNA-seq data. We used deeptools (version 2.29.0) "computeMatrix reference-point" command to separate that distal peak's extended region (defined as a region: 2 kb upstream and downstream from that distal peak center) into 400 10-bp bins and averaging transcription levels of each bin. For each distal peak's extended region, we regarded region 500 bp upstream and downstream as the core peak region while other sites were a nearby region. We used the

Wilcox test and Benjamini–Hochberg *p*-value adjustment method in R to test if the core peak region transcription level was significantly greater than the nearby region (adjusted *p*-value < 0.01). We require both the RPKM value of FIPRESCI data larger than 1 and an adjusted *p*-value < 0.01 for a distal ATAC peak to be called as eRNA locus.

### Analysis of peripheral T cell gene expression profile from FIPRESCI

We used Seurat (version 4.0.5) to analyze the peripheral T cell gene expression profile from FIPRESCI. Firstly, cells with mitochondria ratio > 5% and gene number detected < 50 were filtered. To annotate cell types, we performed label transfer based on a public well-annotated CITE-seq PBMC atlas [27] by Seurat function "FindTransferAnchors" (parameters: normalization.method = "SCT", reference.reduction = "spca", dim = "1:30″) and Seurat function "MapQuery" (parameters: " reduction.model = "wnn.umap", reference.reduction = "spca", refdata = "celltype.l2″). After label transfer, we filtered cells whose prediction score is smaller than 0.4, then we visualized remained cells embedding on reference UMAP by setting a parameter: reduction = "ref.umap" in Seurat function "DimPlot". After annotation and embedding, we calculated each cancer donor's cell type proportion changes compared with healthy donors. To identify differentially expressed genes, we performed a Wilcox test and calculated the average log2 fold change value between two groups of cells using the Seurat function "FindMarkers". To find other differential features, we also imputed surface protein expression levels for our peripheral T cell data based on the CITE-seq PBMC atlas by Seurat function "MapQuery" ( parameters: refdata = list (predicted_ADT = "ADT")).

### Analysis of Treg subpopulations in peripheral T cells

We mainly used Seurat (Version 4.0.5) to analyze Treg subpopulations in peripheral cells. We extracted the Treg expression matrix (UMI count matrix) from the label transferred PBMC matrix described before. Then, we normalized and scaled the gene expression matrix by corresponding Seurat function with default parameters. After PCA, a shared nearest neighbor graph was constructed by the Seurat function "FindNeighbors" with the top 20 PCs. Finally, unsupervised clustering was applied on Treg cells with Louvain algorithm resolution = 0.4, and so we got five Treg subpopulations. We used the R package ComPlexHeatmap [41] to show five subpopulation over donors.

To identify which subpopulation is at an early stage, we implemented CytoTRACE [31] to the Treg gene expression matrix to calculate the CytoTRACE score for each cell. A cell with a higher CytoTRACE score means less differentiated or at an earlier stage.

### Analysis of TCR profile from FIPRESCI

FASTQ files generated from the FIPRESCI TCR profile were used as input for 10X Genomics software Cell Ranger vdj (version 6.0.2) with default parameters. Enriching VDJ by nested PCR eliminates the first round of barcodes introduced by tagmentation. To obtain TCR with single-cell resolution, we only kept the droplets containing one cell (singlet) based on two rounds of barcode information from corresponding scRNA sequencing data. These reads with barcodes must be satisfied the following criteria: (1) Corresponding cells in T cell gene expression profile data must be successfully annotated (annotation procedure was described before); (2) Round 2 barcodes (10X

Genomics cell barcode) must be unique when considering preindexing (round 1) barcodes. After raw sequencing data processing, we used scRepertoire (version 1.3.2) [42] for downstream analysis. Briefly, we calculated unique clonotype numbers by scRepertoire function "quantContig" (parameter: cloneCall = "gene + nt"). We also looked at the length distribution of the CDR3 sequences in the TRA and TRB chain by using the function lengthContig with respective default parameters. Gene usages in both chains were investigated by function vizGenes with respective default parameters. Function clonalProportion helped us to see the top clonotypes (ranked by frequency of occurrence) proportion within one sample. Clonotype diversity was calculated by function clonalDiversity (parameters: cloneCall = "gene + nt", n.boots = 1000 (for Treg subpopulation, n.boots = 100)). Combined with the R package "circlize" (version 0.4.13) [43], we used scRepertoire function getCirclize to visualize shared clonotypes across different groups of cells.

### Co-clustering of peripheral TCR and NeoTCR
We downloaded NeoTCR CDR3 amino acid sequence from recent work [44]. We firstly filter cells that only detected a single chain of CDR3 amino acid sequence, and then we used the software GIANA [45] to cluster our peripheral TCR and NeoTCR CDR3 amino acid sequence (single chains) by default parameters (except –M TRUE –v FALSE). Since GIANA can embed the TCR CDR3 sequence into a 96-dimension space, we calculated pairwise pearson distance in the space. Finally, we visualized the TCR clusters and relationships between TCR clusters by R package ape (Version 5.6–2) function nj with default parameters and the pairwise distance matrix.

### ATAC peak signal prediction model building
Hela FIPRESCI uniquely barcoded cells (UBCs) (for processing easily) were used as training data. The followings are training procedures. Firstly, to equip Hela FIPRESCI cells with ATAC peak signals, we matched Hela FIPRESCI cells with ATAC-seq cells from public Hela scATAC-seq data by OptMatch. We called two cells a cell pair if they were matched cells. A cell pair was regarded as one pseudo cell that had two modalities: RNA profile and ATAC profile. Secondly, to extract eRNA information from FIPRESCI data, FIPRESCI R1 reads whose distance to all TSS was greater than 1 kb or overlapped with any exon were filtered. Remained reads were overlapped with bins from the scATAC-seq bin matrix to get read counts at each peak. Thirdly, we filtered bins occurring in fewer than 0.1% of cells or more than 10% of cells. Bins that passed the filter criterion were used as complements to RNA modality in cell pairs. Finally, we employed BABEL to train an ATAC signal prediction model on cell pairs.

### Computing the correlation of predicted peak signal and bulk peak signal
We used the ATAC signal prediction model described before to predict ATAC signals from FIPRESCI unpaired Hela cells (3154/5539). We can get a cell verse peak probability matrix $P$, $P(i,j)$ indicated peak $i$'s accessible possibility in cell $j$. The sum of the probability values of all cells for each peak was used as the predicted peak signal.

Dividing the whole genome as bins at 500 bp, we applied the same coordinate to scATAC peaks and bulk peaks. To reduce multiple overlaps when coordinate shift,

bedtools was implemented to compute the intersection between predicted peaks and whole genome bins (bedtools -f = 0.5), then we got 327,516 bins as bin set A, while from bulk ATAC-seq peaks and whole genome bins, we got 23,284 bins as bin set B (bedtools -f = 0.2). The common bin number in Set A and Set B is 14,676. After ranking overlapped predicted peak bin signals, we computed the correlation between the top 6000 predicted peak signals and experimental bulk peak signals.

### Analysis of E10.5 mouse embryo gene expression profile

We used Seurat (version 4.0.5) [27] to analyze the E10.5 mouse embryo gene expression profile from FIPRESCI snRNA-seq. Firstly, cells with mitochondria ratio > 5% and gene number detected < 50 were filtered. After normalization and PCA, the top 20 PCs were used to construct a KNN graph by the Seurat function "FindNeighbors". Then, the Louvain algorithm was applied to clustering with resolution = 0.4. Differentially expressed genes for each cluster were identified by the Seurat function "FindAllMarkers" (parameters: only.pos = TRUE) and other default parameters. UMAP was performed for visualization.

For integration analysis, we downloaded a public scRNA-seq profiled mouse organogenesis atlas [24]. For convenient calculation, we used randomly sampled 20,000 annotated E10.5 mouse embryo cells from the atlas as reference. Before label transfer, the normalization method SCtransform [46] was applied to our FIPRESCI data to reduce the sequencing depth's impact. Then, we performed label transfer by Seurat function "FindTransferAnchors" (parameters: normalization.method = "SCT", reference.reduction = "pca", dim = "1:20") and Seurat function "TransferData". After label transfer, we filtered cells whose prediction score is smaller than 0.4.

### Trajectory analysis of E10.5 mouse brain

We mainly used monocle3 (version 1.0.0) [24] to analyze the E10.5 mouse brain trajectory. We extracted brain cells from whole E10.5 mouse embryo FIPRESCI data. Then we implemented Scanpy (version 1.8.2) to construct a PAGA (Partition-based Graph Abstraction) graph. UMAP embedding of those cells was renewed by the Scanpy function "Scanpy.tl.umap" (parameters: init_pos = "paga"). Monocle3 was applied to the renewed UMAP embedding cells. Trajectory was learned by monocle3 function "learn_graph" (parameters: close_loop = FALSE) and other default parameters. With the prior knowledge that Neural Tube cells are the earliest cells, we programmatically determined the root principal node in trajectory: firstly, grouping cells according to which trajectory graph node they are nearest to, then calculating what fraction of the earliest cells (Neural Tube) at each node, node who is most heavily occupied by the earliest cells is selected as root (source code is available at monocle3 tutorial pages). Beginning with the root node, pseudo time was assigned to each cell by the monocle3 function "order_cells" with default settings.

Trajectory graph-correlated genes were identified by the monocle3 function "graph_test" with default settings. Then we selected genes with a *q* value smaller than 0.01 after "graph_test". Gene modules were identified using those genes by the monocle3 function "find_gene_modules" (parameters: resolution = 0.1, partition_qval = 0.01).

Focused on inhibitory neuron trajectory, we selected inhibitory neuron trajectory related trajectory nodes, graphs, and cells to obtain an inhibitory neuron trajectory sub-graph from the whole graph. We also identified trajectory graph-correlated genes by monocle3 function "graph_test". Genes with $q$ value $< 0.05$ and Moran's index $> 0.05$ after "graph_test" are selected as graph-correlated genes. Then $K$-mean clustering ($k = 5$) was performed on those genes (gene verse pseudo time matrix). Finally, we use ComplexHeatmap (version 2.6.2) to show those graph-correlated gene expression's dynamic changes along pseudo time.

### TSS analysis of E10.5 mouse brain from FIPRESCI

We used the Paraclu peak caller [47] to identify TSS. For unsupervised clusters, we first grouped unique mapping reads according to clusters. We call peaks by Paraclu peak caller per cluster. Then we merged peaks from all clusters by bedtools (Version v2.30.0) and filter peaks whose width is greater than 300 bp. Finally, peaks whose distance to any annotated TSS (from mm10 reference genome) is smaller than 100 bp would survive. Those survived peaks are regarded as TSS peaks. Reads mapping those TSS peaks can be regarded using the corresponding TSSs.

For inhibitory neuron trajectory TSS analysis, we first grouped unique mapping reads by equally 10 bins of pseudo time. Then we merged the first 3 bins (in pseudo time increasing order) as the early stage of trajectory, the 4th to 7th bins as the medium stage, and the 8th to 10th bins as the later stage. We used the same peak calling strategy described before, the only difference is we used 3 groups of reads (early, medium, and later) rather than 15 unsupervised clusters.

To control for differences in overall gene expression, we computed the relative usage for each TSS by its read proportion of all TSS reads in the corresponding gene. This usage proportion would be converted to an integer by multiple scale factor 100 and then ceiling to the nearest integer. Within a cluster, we treated a gene's TSS read count value as missing if the sum of its TSSs' reads is smaller than 0.5 quantile or greater than 0.99 quantiles over the cluster's gene TSSs' read count distribution. We compared the relative usage for the set of TSSs in a given gene between each pair of samples using Fisher's exact test implemented using the R function fisher.test. The test was performed on the 2 (samples used for comparison) by $n$ (TSSs within one gene) table. We then took the minimum Benjamini-Hochberg-corrected $p$-value across all comparisons (for the multiple comparisons per gene), reported as the $p$-value for differential usage across the samples within a gene. Within one TSS, the variance between the 2 samples that used the top 2 most proportion of that TSS was reported as the variance of the TSS.

## Supplementary Information

**Additional file 1: Figure S1-S13.**

**Additional file 2: Tables S1.** Oligonucleotide sequences for Fipresci-Seq.

**Additional file 3: Tables S2.** Statistics of Fipresci-Seq Round1 index oligonucleotide sequences used for all experiments in this study.

**Additional file 4: Tables S3.** List of the component of all Tn5 tagmentation reaction buffer conditions.

**Additional file 5: Tables S4.** Statistics of PBMC donor's information.

---

**Additional file 6.** Legends and web links for processed files S1-S12.

**Additional file 7.** Review history.

---

## Acknowledgements
We thank the members of the Jiang lab for helpful discussions. We thank Dr. Yi Zhang and Mr. Qiangzong Yin from Harvard Medical School for the early exploratory work of low-input and single tube 5'-end RNA-seq back in 2018.

## Review history
The review history is available as Additional file 7.

## Peer review information

## Authors' contributions
L.J. conceived the study. L.J. and Y.L. developed the FIPRESCI methods. Y.L. performed the experiments. Z.H. performed the bioinformatics analyses. Q.W. performed computational modeling. Z.Z. facilitated the experiments of primary T cells of human cancer. S.W., X.F, and F.L. collected the human peripheral blood. S.S. and X.J. contributed to the discussion and provided critical review and/or revision of the manuscript. Y.L. and L.J. wrote the manuscript with the assistance of the other authors. The authors read and approved the final manuscript.

## Authors' Twitter handles
Twitter handle: @LanJiangBoston (Lan Jiang).

## Funding
L.J. was supported by the Strategic Priority Research Program of the Chinese Academy of Sciences (XDB38020500), the National Key Research and Development Program of China (2019YFA0801702), the National Natural Science Foundation of China (31970760), and the International Partnership Program of the Chinese Academy of Sciences (153F11KYSB20210006). X.F. was supported by the Strategic Priority Research Program of the Chinese Academy of Sciences (No. XDA16010602), and the National Natural Science Foundation of China (No. 82070114, 81870097, 32071117).

## Availability of data and materials
The raw sequence data reported in this paper have been deposited in the Genome Sequence Archive in the National Genomics Data Center, Beijing Institute of Genomics (China National Center for Bioinformation) of CAS under accession No. PRJCA007160 at https://ngdc.cncb.ac.cn/gsa/ [48]. All codes used to analyze data are available on GitHub, at https://github.com/hz1010/FIPRESCI_data_analysis [49] and Zenodo https://zenodo.org/badge/latestdoi/603282281 [50]. Legends and web links for processed files S1-S12 can be found in Supplementary file 1.

## Declarations

### Ethics approval and consent to participate
The study was approved by the Ethics Committee of Chinese PLA General Hospital (S2022-643–01) and was conducted following the Declaration of Helsinki. All the patients in this study have provided written informed consent.

### Competing interests
L. J. and Y.L. are inventors of a patent application describing the FIPRESCI method. The other authors declare that they have no competing interests. The patent does not restrict the use of the method for educational, research, or not-for-profit purposes.

### Author details
[1]China National Center for Bioinformation, Beijing 100101, China. [2]CAS Key Laboratory of Genome Sciences and Information, Beijing Institute of Genomics, Chinese Academy of Sciences, Beijing 100101, China. [3]University of Chinese Academy of Sciences, Beijing 100049, China. [4]Sino-Danish College, University of Chinese Academy of Sciences, Beijing 100049, China. [5]The Blood Transfusion Department, First Medical Center of Chinese, PLA General Hospital, Beijing 100853, China. [6]Key Laboratory of Carcinogenesis and Translational Research (Ministry of Education/Beijing), Gastrointestinal Cancer Center, Peking University Cancer Hospital & Institute, No. 52 Fucheng Road, Beijing 100142, China. [7]Peking University International Cancer Institute & Peking University Cancer Hospital & Institute, Beijing 100191, China. [8]Institute for Stem Cell and Regeneration, Chinese Academy of Sciences, Beijing 100101, China. [9]Beijing Key Laboratory of Genome and Precision Medicine Technologies, Beijing 100101, China. [10]College of Future Technology College, University of Chinese Academy of Sciences, Beijing 100049, China.

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

## 