## [**Additional file 7.** Review history. · Genome Biology]

Review History

First round of review

Reviewer 1

Were you able to assess all statistics in the manuscript, including the appropriateness of statistical tests used? No.

Were you able to directly test the methods? No.

Comments to author:

In the manuscript entitled "Droplet microfluidics based combinatorial indexing for massive-scale 5'-end single-cell RNA sequencing" Li et al. present a new method to increase the cell throughput of commercial single cell microfluidics instruments like the 10X Chromium Controller. The work is similar in concept to the cited sci-fi-RNA-seq method where in an initial plate based step, cells are split and barcoded before loading them onto the 10X Controller for further partitioning and barcoding. The FIPRESCI approach presented by the author is original in using the tagmentation of cDNA/RNA hybrids to make it possible to capture the 5' end of RNA molecules instead of the 3' end like other approaches. The authors describe how that makes it possible to investigate TSSs and TCR VDJ sequences.

While there clearly is promise in the approach for high throughput single cell analysis, there are several major issues with the work as presented:

1) The by far largest concern that I have is that the authors describe how the FIPRESCI approach is compatible with 10X Genomics protocols for the sequencing of T cell receptors (TCRs). This involves performing two rounds of nested PCR with primers specific to the TCR constant region. While this works on regular 10X libraries which are only barcoded by the TSO, here it would very much seem like the nested PCR would eliminate the first-round barcode introduced by tagmentation. That would make it impossible to assign TCR VDJ sequencing reads to individual cells. Therefore I don't see how the analysis in the paper could have been performed as described.

A lot more detail would have to be provided on how this analysis was done to explain how and why it was possible to do so based on the barcoding scheme that was used.

2) Based on the data provided by the authors, the predictive power to identify enhancer loci based on FIPRESCI data is weak. With a Pearson r of 0.33 and Fig 2 f basically being an uncorrelated cloud of points, I don't think FIPRESCI is actually able to identify cis-regulatory elements in any useful way.

3) The structure of the paper makes it hard-to-impossible to know what conditions were used for individual experiments.

Basically, what buffer system and conditions were used in the validation experiment, the optimization experiments, the embryo analysis and the T cell analysis. The methods section suggests that it was "commercial buffer 2" (Vazyme) but the optimization states that

"commercial buffer 1" showed the highest activity. If different, experimental conditions have to be outlined for each separate condition.

4) Several figure panels have missing (e.g. 2c,d) or meaningless (2a "Hela_Expression levels") y-axis labels.

5) Several key stats are missing. In particular, since Tn5 will likely also cut dsDNA in the cell, it would be important to know what percentage of reads falls within annotated gene bodies and how that compares to standard 10X 5' approaches. Also, the insert length of reads would be important to know and whether the sequence read yield on each Illumina flow cell was affected by the presence of p7-p7 molecules.

Reviewer 2

Were you able to assess all statistics in the manuscript, including the appropriateness of statistical tests used? Yes: provide more detail statistics and methods for calling enhancers based on 5' RNA seq data, what is the cutoff and background noise level to call significant eRNA based enhancer? What is the number of eRNA based enhancer as compared to ATAC and how well is this conserved across different conditions (e.g. poly dT, oligo, mix, fixed, non-fixed, tissues, etc).

Were you able to directly test the methods? No.

Comments to author:

In this manuscript, Li, Huang, Zhang and coauthors described an ultra-high throughput 5'-end single cell combinatorial indexing RNA sequencing protocol that lowers the cost per cell. This protocol uses Tn5 transposome to cut and tag RNA/cDNA hybrids for the first round of barcoding, followed by overloading of cells into droplets for a second round of barcoding. The authors proceeded to apply this protocol on mouse whole embryo and human donor T cells and demonstrated similar performance in gene and cis-regulatory element detection with current 5' end sequencing protocols.

Overall, the writing is clear, but there are a number of spelling errors, some of which are listed below.

1. Page 4 Line 5 - "cell tying" should be "cell typing"
2. Page 4 Line 8 - "inter cis-gene" should be "infer cis-gene"
3. Page 4 Line 30 - "removal of droplets" should be "removal of doublets"
4. Page 5 Line 6 - "throughout" should be "throughput"
5. Page 10 Line 37 - "maker" should be "marker"
6. Page 10 Line 47 - "donnors" should be "donor"

1. There is a chance that Tn5 may ligate the sequence to the RNA strand instead of cDNA strand on the 5' end fragment. If this happens at random, it will mean that 50% of the transcripts will not be captured by FIPRESCI. Although the authors have shown that similar number of genes can be detected, how do the UMI counts compare between FIPRESCI and conventional 5' end scRNA seq?

2. Please describe the template switching step in Supp Figure 2 diagram where transcript extension is done through the TSO but also at the 3' end.

3. What is the frequency to which Tn5 cuts short RNAs such as eRNAs? How does this compared to the conventional 5' end scRNA seq that utilizes fragmentation method?
4. What is the rate of strand invasion, and relatedly, what is the general noise (background) level across non-5' end regions (e.g. exons) as compared to gene expression. Please provide the scale on IGV track found in Figure 3C. The peaks for clusters 7 and 9 look rather low and noisy - clear TSS are not observed. In fact, some of the peaks in these clusters map to intronic regions of annotated transcripts.
5. The authors compared the use of oligo d(T), random and mixed primers in FIPRESCI by looking at the number of detected genes per cell. What is the gene profile and gene detection rate of oligo d(T), mixed and random primer FIPRESCI datasets? Do the different primers enrich for different classes of RNAs? Are the different primer sets suitable for different types of analyses?
6. In this protocol, cells are permeabilized and fixed with methanol. Previous studies have shown that methanol fixation is harsh on RNA and affects reverse transcription, especially with protocols requiring template switching (1, 2). Have the authors compared FIPRESCI with and without methanol fixation, or with formaldehyde fixation?
7. FIPRESCI was performed on human donor T cells and annotated using a CITE-seq reference dataset. However, the purpose of imputing protein expression levels using this dataset is unclear. 7 T cell clusters could already be identified and annotated using gene expression data. Protein expression was imputed from gene expression and did not improve the clustering. Can the authors elaborate further on the additional insights from imputing protein expression?
8. Since Tn5 exhibits a motif and shape bias when binding dsDNA (3), does Tn5 also have a sequence bias when binding DNA/RNA hybrids?

1. Phan HV, van Gent M, Drayman N, Basu A, Gack MU, Tay S. High-throughput RNA sequencing of paraformaldehyde-fixed single cells. *Nature Communications*. 2021;12(1):5636.
2. Wang X, Yu L, Wu AR. The effect of methanol fixation on single-cell RNA sequencing data. *BMC Genomics*. 2021;22(1):420.
3. Zhang H, Lu T, Liu S, Yang J, Sun G, Cheng T, et al. Comprehensive understanding of Tn5 insertion preference improves transcription regulatory element identification. *NAR Genomics and Bioinformatics*. 2021;3(4):lqab094.

Authors Response

We thank the reviewers for their critical assessment of our manuscript and for the suggestions to improve this effort. We have given their feedback careful thought and revised our manuscript and analyses to reflect their suggestions. A point-by-point response detailing what we have changed as well as explanations to support our claims, is presented below, in blue.

Reviewer reports:

Reviewer #1: In the manuscript entitled "Droplet microfluidics based combinatorial indexing for massive-scale 5'-end single-cell RNA sequencing" Li et al. present a new method to increase the cell throughput of commercial single cell microfluidics instruments like the 10X Chromium Controller. The work is similar in concept to the cited sci-fi-RNA-seq method where in an initial plate based step, cells are split and barcoded before loading them onto the 10X Controller for

further partitioning and barcoding. The FIPRESKI approach presented by the author is original in using the tagmentation of cDNA/RNA hybrids to make it possible to capture the 5' end of RNA molecules instead of the 3' end like other approaches. The authors describe how that makes it possible to investigate TSSs and TCR VDJ sequences.

While there clearly is promise in the approach for high throughput single cell analysis, there are several major issues with the work as presented:

1) The by far largest concern that I have is that the authors describe how the FIPRESKI approach is compatible with 10X Genomics protocols for the sequencing of T cell receptors (TCRs). This involves performing two rounds of nested PCR with primers specific to the TCR constant region. While this works on regular 10X libraries which are only barcoded by the TSO, here it would very much seem like the nested PCR would eliminate the first-round barcode introduced by tagmentation. That would make it impossible to assign TCR VDJ sequencing reads to individual cells. Therefore I don't see how the analysis in the paper could have been performed as described. A lot more detail would have to be provided on how this analysis was done to explain how and why it was possible to do so based on the barcoding scheme that was used.

Answer: We thank the reviewer for raising this important point. We acknowledge that our method about high throughput single-cell TCR relevant analysis was not properly documented. We agree that enriching VDJ by nested PCR like conventional 10x Genomics does eliminate the first round of barcodes introduced by tagmentation in FIPRESKI approach. However, we still are able to obtain single-cell resolution TCR information, just with a relative low recovery rate.

The cell barcode in FIPRESKI data is defined by Tn5 introduced 1st round barcode and droplet introduced 2ed round barcode. We identified the droplets containing a single cell as “singlet” and droplets containing more than one cell as “multiplet” based on the two rounds of barcode. For transcriptome and TSSs analysis, we include all data from singlets and multiplets. While for VDJ analysis, we exclude the data from multiplets. In other words, for singlet, the droplet introduced 2ed round barcode alone can clearly define the cell barcode for both transcriptome and VDJ data.

We want to emphasize that, the proportion of useful sequencing data of FIPRESKI is higher than 90%, since the transcriptome from multiplets can be resolved and VDJ libraries only account for a very small amount of whole dataset. In addition, although with the limitation in terms of the low recovery rate of VDJ sequencing, FIPRESKI is still a valuable approach for TCR analysis as we have shown that it is able to reveal the “TCR clonotype diversity” and “shared TCR clonotypes among cell types and across donors” due to the large number of total cells (Fig 5 and Supplementary Figure 12, 13).

We have mentioned “Analysis of TCR profile from FIPRESKI” in Methods section in the previous version (Page 23, line 51). To address this problem, we have re-described it in the Results (Page 9, line 44), and Methods (Page 21, line 28) of the revised version of the manuscript.

2) Based on the data provided by the authors, the predictive power to identify enhancer loci based on FIPRESKI data is weak. With a pearson r of 0.33 and Fig 2 f basically being an uncorrelated cloud of points, I don't think FIPRESKI is actually able to identify cis-regulatory elements in any useful way.

Answer: We thank the reviewer for raising this important point and apologize for not presenting our results well in the previous manuscript. This part is actually related to two different topics: 1) using 5'-end RNA-seq data to identify distal transcribed cis-regulatory elements or enhancer derived RNAs(eRNAs); 2) using RNA-seq data to predict ATAC-seq signal at single peak level. Although ATAC-seq is broadly accepted to identify cis-regulatory elements, ATAC-seq alone cannot tell which distal open chromatin region is actively transcribed (eRNA). For the first topic, Drs. Jay W. Shin and Chung-Chau Hon have recently developed an elegant computational method called SCAFE (<https://www.biorxiv.org/content/10.1101/2021.04.04.438388v2>) to address this need, which is cited in our previous manuscript. In addition, if matched ATAC-seq data is available for the same sample or same cell type, it is straightforward to identify eRNA transcription start sites (eTSSs) by combing 5'-end RNA-seq data and ATAC-seq data. We have applied this approach and reanalysis our FIPERSCI-seq data in Fig 2. If a distal ATAC peak called by MACS is enriched with strong FIPERSCI-seq signal (RPM >0.5), we defined it as an eRNA locus. We found FIPERSCI can identify 4015~4913 eRNA loci, which account for 34.4%~42.1% distal ATAC peaks. Furthermore, in panel C, the eRNA profile among different conditions is similar, which indicated our eRNA detection is robust. We want to note that it is not unusual both of ATAC-seq data and 5'-end RNA-seq data are available for a given sample or cell type. For example, we have applied 5'-end scRNA-seq and scATAC-seq to 4 human fetal embryos including 15 major organs, and revealed more than 200 cell types and ~800K open chromatin regions (<https://www.biorxiv.org/content/10.1101/2021.11.02.466852v1>). However, this work is based on standard 10X Genomics procedure, which is cost prohibitive. Ultra-high-throughput methods, like dsciATAC-seq and FIPERSCI, will facilitate those large-scale projects.

For the second topic, using RNA-seq signal to predict ATAC-seq signal is indeed a challenging task. BABEL can predict ATAC-seq signals (cell vs. peak matrix) from RNA-seq data (Cell vs. gene matrix). The previously Fig 2 e,f,g is trying to illustrate that when using BABEL model to predict ATAC-seq signal, using 5'-end RNA-seq has additional benefit compared to 3'-end RNA-seq because 5'-end RNA-seq have enriched signal centered with enhancer(Fig 2d). We agreed this part is premature and confusing. Thus we have moved previous Fig 2 e,d,f to supplemental results and replaced them with eRNA identification results mentioned above. To address the reviewer's concern, we also rewrite the relevant part in Main text (line 19 on Page 6).

3) The structure of the paper makes it hard-to-impossible to know what conditions were used for individual experiments.

Basically, what buffer system and conditions were used in the validation experiment, the optimization experiments, the embryo analysis and the T cell analysis. The methods section suggests that it was "commercial buffer 2" (Vazyme) but the optimization states that "commercial buffer 1" showed the highest activity. If different, experimental conditions have to be outlined for each separate condition.

Answer: We appreciate the reviewer's comment. We apologize for not describing clearly which buffer systems and conditions are used. The different conditions involved in individual experiments include TN5 tagmentation buffer and RT primers. First, for TN5 tagmentation buffer condition, we used "commercial buffer 2 (Vazyme# S601-01)" in validation experiments. We then tested 16 tagmentation buffer conditions (the detailed components and performance of tagmentation buffers are provided in Supplementary Table 3 and Fig. 2a, and Supplementary Fig. 4b) in the "optimization experiment 1" and found that "commercial buffer 1(Apexbio #K1155-20ul)" showed the highest activity. Unfortunately, the supply of "commercial buffer 1" is disrupted during our project. So, for the rest of the subsequent experiments we used the customized tagmentation buffer (10 mM Tris-HCl at pH 7.5, 5 mM MgCl₂, and 10% DMF) which had the second highest activity performance in the optimization experiment. Second, for the RT primers condition, only "optimization experiment 2" involved the use of random hexamer primer, and a mix of oligo d(T) and random primers, all other experiments used oligo d(T) primers only. In the updated manuscript, we have followed the reviewer's comments and modified the description in the Results (Page 5, line 26) and Methods sections (Page 14, line 26). We have summarized the reaction conditions, as well as additional information for each individual experiment below and also in Supplementary Table 3.

4) Several figure panels have missing (e.g. 2c,d) or meaningless (2a "Hela Expression levels") y-axis labels.

Answer: Thank you very much for pointing out those errors. We have corrected them in the revised version and also shown below.

5) Several key stats are missing. In particular, since Tn5 will likely also cut dsDNA in the cell, it would be important to know what percentage of reads falls within annotated gene bodies and how that compares to standard 10X 5' approaches. Also, the insert length of reads would be important to know and whether the sequence read yield on each Illumina flow cell was affected by the presence of p7-p7 molecules.

Answer: Thank you for the constructive comments and valuable suggestions. To avoid p7-p7 molecules, we have used biotinylated P5 primers to specifically enrich for 5'-end RNA fragments, which is mentioned in Supplementary Table 1 of the previous manuscript. We apologize for forgetting to describe this detail in the Methods. Following your suggestion, now we have modified the cDNA enrichment part in the Methods section (line 23 on Page 15) and updated more details in Supplementary Figure 2.

The percentage of reads fall in gene body in our data is shown in panel a below, which is

comparable with the conventional 10X 5' approach (ESCC data from PMID: 34489433; Mouse embryo data from our unpublished work). The results support that the reads in our data are less likely from dsDNA. The distribution of insert size is shown in panel b, which clearly exhibits a pattern different from ATAC-seq data.

Reviewer #2: In this manuscript, Li, Huang, Zhang and coauthors described an ultra-high throughput 5'-end single cell combinatorial indexing RNA sequencing protocol that lowers the cost per cell. This protocol uses Tn5 transposome to cut and tag RNA/cDNA hybrids for the first round of barcoding, followed by overloading of cells into droplets for a second round of barcoding. The authors proceeded to apply this protocol on mouse whole embryo and human donor T cells and demonstrated similar performance in gene and cis-regulatory element detection with current 5' end sequencing protocols.

Overall, the writing is clear, but there are a number of spelling errors, some of which are listed below.

1. Page 4 Line 5 - "cell tying" should be "cell typing"
2. Page 4 Line 8 - "inter cis-gene" should be "intra cis-gene"
3. Page 4 Line 30 - "removal of droplets" should be "removal of doublets"
4. Page 5 Line 6 - "throughout" should be "throughput"
5. Page 10 Line 37 - "maker" should be "marker"
6. Page 10 Line 47 - "donnors" should be "donor"

Answer: We apologize for those mistakes, which have been corrected in the updated version. We have also carefully reexamined spelling throughout the text.

1. *There is a chance that Tn5 may ligate the sequence to the RNA strand instead of cDNA*

strand on the 5' end fragment. If this happens at random, it will mean that 50% of the transcripts will not be captured by FIPRESCI. Although the authors have shown that similar number of genes can be detected, how do the UMI counts compare between FIPRESCI and conventional 5' end scRNA seq?

Answer: We thank the reviewer for raising this important point. Firstly, we use i7-only transposomes (Tn5-S7/S7 homodimer), thus the products from tagmentation is less complicated. Oligonucleotides Tn5-top_ME and Tn5-bottom_Read2N were synthesized for transposome complex assembly and sequences are provided in Supplementary Table S1. Second, based on the description from the literature (PMID: 12603728, 17693501, 21143862), Tn5 transposase activity will result in fragmentation and end-joining of the Tn5-bottom_Read2N oligo to the 5' end of upper strand of DNA/RNA hybrid. Thus, the probability of Tn5 sequence ligation to the cDNA strand of 5' end fragment (highlighted in red box) should be 100% as shown in Panel A. The Tn5 ligate the sequence to the RNA strand of 5' end fragment may occur only when oligonucleotides Tn5-top_ME_Read2N and Tn5-bottom_ME are used for transposomes assembly as shown in Panel B. However, it is not the case in our experiment.

For the second question, as shown in Panel C, we agree with the reviewer that the UMI counts of FIPERSCI is expected to be lower than the conventional 5' method (data from PMID: 34489433). Probably because it is difficult for the actual efficiency of Tn5 tagmentation on RNA/DNA hybrids to reach 100%. However, we have confirmed that the gene number is comparable (Panel D). In other words, the conversion ratio of UMI to gene is much higher in our data than conventional 5' method. We suspect the fixing before RT procedure of FIPERSCI may help to keep lowly expressed genes from RNA degradation.

c

d

2. Please describe the template switching step in Supp Figure 2 diagram where transcript extension is done through the TSO but also at the 3' end.

Answer: We thank the reviewer for the nice suggestion. To address the reviewer's concern, we have modified the diagram and updated more details in Supplementary Figure 2.

Design for Fipersci-Seq
(by Yun Li, 2020-04-15)

✳ **in situ RT**

(1) in situ Reverse transcription with Poly-dT RT primer using MMLV:

(2) The terminal transferase activity of MMLV adds extra Cs:

✳ **in situ I7 only Tn5 transposition**

(3) Tn5 s7 homodimer to cut the RT product (DNA/RNA hybrid):

✳ **Template switching in 10X Chromium GEM-RT Incubation**

(4) cDNA capture by gel bead barcoded TSO and cDNA extension (in 10X Chromium Droplet):

(5) Break emulsion and purify the products:

✳ **Pre-Amplification**

(6) cDNA enrichment by biotinylated single primer linear amplification:

✳ **Library-amplification**

(7) Add partial p5 primer and I7 Sample Index primer for library amplification:

✳ **Final Library**

3. What is the frequency to which Tn5 cuts short RNAs such as eRNAs? How does this

compared to the conventional 5' end scRNA seq that utilizes fragmentation method?

Answer: We thank the reviewer for raising this important point. As mentioned before, currently we identify eRNA with the help of matched ATAC data. We have found a publicly available conventional 5' end scRNA-seq dataset of ESCC (Esophageal squamous-cell carcinoma) with matched bulk ATAC-seq data. We estimate the contribution of eRNA reads by counting 5' end scRNA-seq reads which fall into ATAC-seq defined distal open chromatin regions (data from PMID: 34722266). As shown in the figure below, generally, the contribution of eRNAs to the whole dataset is low (0.25 % to 2%). This observation is consistent with the notion that eRNAs are short and unstable. The difference between FIPRESKI and conventional 5' end scRNA-seq is significant. However, the samples and batches differences should also be considered.

4. What is the rate of strand invasion, and relatedly, what is the general noise (background) level across non-5' end regions (e.g. exons) as compared to gene expression. Please provide the scale on IGV track found in Figure 3C. The peaks for clusters 7 and 9 look rather low and noisy - clear TSS are not observed. In fact, some of the peaks in these clusters map to intronic regions of annotated transcripts.

Answer: We thank the reviewer for raising those important points. We apologize that the data scale on IGV track in Figure 3C is missing. We have updated Figure 3C and add data scale on the right side of all the IGV track in the manuscript and this response letter. We believe those small peaks from non-5' end regions in cluster 7 and cluster 9 are not noise. Because if those signals are noise generated from mechanism like strand invasion, one would expect to observe similar noise level across the whole genome. However, for many genes, we observed universal extremely low noise ratio across all the clusters. IGV tracks of Hit1 and trim17 are shown as representative examples

(panel a and b). In other word, the “noise” levels are gene specific. Thus, we suspect the “noise” level may related to the stability of particular genes’ promoter or RNA modification of 5’-end, or even varies for different promoters of the same genes. When small peaks from non-5’ end regions of a gene are observed, it is more likely the 5’-end of the transcript is degenerated faster than other genes or promoters, other than a general noise come from FIPRESCI procedure. We have calculated the observed FIPRESCI signal in TSS and gene region vs. random region from whole genome, the fold is 13.38 and 2.0 respectively (panel c). This result is consistng with our previous claim that FIPRESCI signal is highly enriched in transcription start region.

5. The authors compared the use of oligo d(T), random and mixed primers in FIPRESCI by looking at the number of detected genes per cell. What is the gene profile and gene detection rate of oligo d(T), mixed and random primer FIPRESCI datasets? Do the different primers enrich for different classes of RNAs? Are the different primer sets suitable for different types of analyses?

Answer: We thank the reviewer for raising those important points. To address the reviewer’s question, we have performed the analysis. The result indicates that PolyT primers mainly capture protein coding genes as expected. Interestingly, the proportion of rRNA from random primers is only slightly higher than PolyT primers. We didn’t include ribosome RNA depletion step in our analysis, therefore we suspect some specific feature of rRNA may inhibit the Tn5 tagmentation. We conclude that although with slightly differences, the sensitivity of detecting of different classes

of RNAs are similar for oligo d(T), random and mixed primers.

6. *In this protocol, cells are permeabilized and fixed with methanol. Previous studies have shown that methanol fixation is harsh on RNA and affects reverse transcription, especially with protocols requiring template switching (1, 2). Have the authors compared FIPRESCI with and without methanol fixation, or with formaldehyde fixation?*

Answer: We thank the reviewer for raising this important point. Because FIPRESCI is a combinatorial indexing approach which require the cells or nuclei to remain intact before being encapsulated in droplets, we have not tested FIPRESCI without any fixation. However, in a separate study, we have compared standard 10x Genomics scRNA-seq (only droplet barcoding, without combinatorial indexing) with and without methanol fixation. Surprisingly, at the same sequencing depth (50K reads/cell), methanol-fixed PBMC achieved a higher median gene per cell compared to fresh (without methanol fixation) PBMC (as following figure shows). We therefore speculate that methanol fixation may inactivate endogenous RNases thus contribute significantly to the preservation of RNA before RT is complete. Regarding fixing with formaldehyde, we have not compared formaldehyde v.s. methanol fixed cell for FIPRESCI, but we have done this comparison for nuclei based Multiome analysis. The results show methanol can recover more genes. We suspect that formaldehyde fixing may require reverse cross link thus less likely to be a better condition than methanol fixation, or need sustainably testing to find the optimal concentration. We agree with the reviewer that we cannot rule out the possibility methanol is not the best fixed condition. However, comprehensively testing all condition is out of the scope of this work. And so far, methanol fixation works pretty well in our hand.

7. FIPRESICI was performed on human donor T cells and annotated using a CITE-seq reference dataset. However, the purpose of imputing protein expression levels using this dataset is unclear. 7 T cell clusters could already be identified and annotated using gene expression data. Protein expression was imputed from gene expression and did not improve the clustering. Can the authors elaborate further on the additional insights from imputing protein expression?

Answer: We highly appreciate the reviewer's comment. We agree with the reviewer that the protein expression imputation has provided limited additional insights so far. And we apologize for not introducing well the background and motivation of this analysis. It is another angle to show the value of FIPRESICI approach. As the multimodality methods, like CITE-seq, usually have limitations about scalability, it is still difficult to generate a large-scale dataset based on CITE-seq. Thus, it may be a good idea to generate a large-scale multimodality dataset based on three things: 1) generating a large-scale dataset of single-modality with a cost-effective method 2) generating a small-scale multimodality reference dataset 3) building a precise imputation method. Thus, when multimodality reference dataset and computational imputation method are available, FIPRESICI will be a powerful approach for this type of task as it's an ultra-high-throughput and cost-effective scRNA-seq method.

8. Since Tn5 exhibits a motif and shape bias when binding dsDNA (3), does Tn5 also have a sequence bias when binding DNA/RNA hybrids?

Answer: We appreciate the reviewer's comment. To address the reviewer's question, we have performed motif analysis. The result (panel a) indicates that Tn5 has a sequence bias when binding DNA/RNA hybrids and the motif is similar to binding dsDNA (based on ATAC-seq from literature, panel b). But the motif signal is weaker than binding dsDNA.

Additional comments:

-provide more detail statistics and methods for calling enhancers based on 5' RNA seq data, what is the cutoff and background noise level to call significant eRNA based enhancer? What is the number of eRNA based enhancer as compared to ATAC and how well is this conserved across different conditions (e.g. poly dT, oligo, mix, fixed, non-fixed, tissues, etc).

Answer: We appreciate the reviewer's comment. As described in the answer of question 2 from review#1, if a distal ATAC peak called by MACS is enriched with strong FIPERSCI-seq signal (RPM >0.5), we defined it as an eRNA locus. We found FIPERSCI can identify 4015~4913 eRNA loci, which account for 34.4%~42.1% distal ATAC peaks. Furthermore, in panel C, the eRNA profile among different conditions is similar, which indicated our eRNA detection is robust. In the updated manuscript, we have followed the reviewer's comments and updated the Methods sections (line 28 on Page 20) to provide more detail statistics and methods for calling enhancers based on 5' RNA seq data.

-Figure 4 panel a, UMAP looks unusually biased to highlight major clusters (cells in the upper left corner are missing?)

Answer: We appreciate the reviewer's comment. There are several small cell clusters in the upper left corner of the original UMAP diagram (shown below). Since these clusters contain less than 10 cells and are not annotated as T cells, we speculate they are FACS errors. Therefore, we have labeled these cells as others in the Figure 4 panel a.

1. Phan HV, van Gent M, Drayman N, Basu A, Gack MU, Tay S. High-throughput RNA sequencing of paraformaldehyde-fixed single cells. *Nature Communications*. 2021;12(1):5636.
2. Wang X, Yu L, Wu AR. The effect of methanol fixation on single-cell RNA sequencing data. *BMC Genomics*. 2021;22(1):420.
3. Zhang H, Lu T, Liu S, Yang J, Sun G, Cheng T, et al. Comprehensive understanding of Tn5 insertion preference improves transcription regulatory element identification. *NAR Genomics and Bioinformatics*. 2021;3(4):lqab09

Second round of review

Reviewer 1

In their point-by-point responses and the revised manuscript, the author did address my main concerns.

The questions that I had about the TCR portion of the manuscript have been cleared up and the methods have been updated enough to make it clear how experiments were performed.

In its current shape, the manuscript should serve as an interesting addition to the expanding literature of pushing the throughput of single cell genomics assays.

Reviewer 2

The revised version raised additional key concerns. Please address the following questions based on previous inquiries.

3. The eRNA capture ratio of 0.025 to 0.02 seems to contradict the eRNA number/distal peak in Figure 2 where it is 0.3 to 0.4. Please explain the differences in these two methods and why one contradicts the other. Is Figure Can the authors show the eRNA insert size profile of the FIPRESCI libraries compared to conventional 5' end scRNA seq eRNAs? This will give an idea of the minimum eRNA length tagmented by Tn5 compared to fragmentation.

5. The ratio of UMIs for rRNA is surprisingly low for Random as compared to Mix and PolyT. The authors suggest that this feature may be due to rRNA inhibiting Tn5 tagmentation. Is there supporting evidence or literature to claim this? Further, rRNAs are abundantly expressed than any other protein coding genes so the ratio of UMI should also be relatively high. Have authors compared the percentage of detected promoter types, where it should better reflect the diversity of genes that this method can detect (instead of gene expression ratios). Many of lncRNAs are polyA minus but this analysis seems to suggest that lncRNA detection is equal across all conditions. Which lncRNA annotation was used for this analysis? It will be useful to include the analysis results in the revised manuscript.

6. The violin plot illustrates a strange pattern for fixed condition where the gene detection and UMI are bimodally distributed as compared to fresh condition. This seems to suggest that there is a group of genes that are more amendable to fixation conditions (e.g. possibly due to RNA structure, bound proteins, transcript length and GC content

<https://bmcbgenomics.biomedcentral.com/articles/10.1186/s12864-021-07744-6>) than others and cause biases in gene detection and expression levels due to fixation method. This can lead to spurious read counts – especially at the 5' end - thus confounding the conclusions made in this paper. Resolving this issue, e.g. what is the cause of this bias and addressing them will be absolutely critical to evaluate the conclusions made in this manuscript.

7. Although protein expression imputation would be useful for the reasons mentioned by the authors, it can be implemented on all single cell RNA-seq datasets in general, and the authors have not shown any particular advantage in performing this analysis with the FIPRESCI dataset. The motivation for protein expression imputation and these results should be placed in the discussion section as a potential application for FIPRESCI.

Additional comments:

While the detection of eRNAs under the pretext that they are within ATAC peaks is notable, there are ample spurious reads that are mapped across the human genome derived from single cell RNA-seq methods including 10x 5'. Therefore it will be imperative for authors to demonstrate that these eRNAs are significantly more enriched than reads that are outside of ATAC peaks, plus the drop off rates based on varying degree of threshold set for RPM as 0.5 is arbitrary and difficult to grasp the sensitivity and accuracy of this level. The response from authors state that on page 20 line 28 provides more detail statistics and methods but the revised version lacks description and should provide a more detailed statistics the claim that these eRNAs are in fact true and not derived from random/spurious mapped reads. Furthermore, eRNAs exhibit bidirectional transcription. Please demonstrate that these eRNAs are divergently transcribed at the single base resolution derived from TSS (e.g. in Figure 2d) and that they largely overlap with canonical histone markers including H3K27ac.

Point-to-point response to the reviewer's comments

Reviewer #2

The revised version raised additional key concerns. Please address the following questions based on previous inquiries.

3. The eRNA capture ratio of 0.025 to 0.02 seems to contradict the eRNA number/distal peak in Figure 2 where it is 0.3 to 0.4. Please explain the differences in these two methods and why one contradicts the other. Is Figure Can the authors show the eRNA insert size profile of the FIPRESCI libraries compared to conventional 5' end scRNA seq eRNAs? This will give an idea of the minimum eRNA length tagged by Tn5 compared to fragmentation.

Answer: Thanks for the reviewer's comment. "The eRNAs capture ratio of 0.025 to 0.02" indicates the vast majority of reads, which are generated from Tn5 cutting on cDNA/RNA hybrid, are from protein-coding genes. The results are calculated by #reads of 5'-end RNA-seq which overlap with distal ATAC peaks defined enhancer region/total number of reads. "eRNA number/distal peaks in Figure 2 where it is 0.3 to 0.4" indicates a significant portion of ATAC data defined enhancer or distal ATAC peak have strong eRNAs signal. The result is calculated by # of distal ATAC peaks which overlap with eRNAs/total number of distal ATAC peaks. The difference is because the calculation is different and consistent with the fact that most of the RNA-seq reads are not overlapped with distal ATAC peaks. We have added the eRNA capture ratio as Supplementary Fig. S5c and updated the label and legend of Figure 2 to make it clearer.

As suggested by the reviewer, we have plotted the eRNA insert size below (R2 Figure 1a). The minimum eRNA length is about 80 nt (R2 Figure 1b). It is worth to note this size distribution is consistent with the insert size profile of whole FIPRESCI libraries when we addressed reviewer #1's question 5. The length of paired-end reads may not reflect the real size from eRNA but simply reflect the result of size selection of the experimental procedure. We have added the following sentence to the discussion (Page 11, Line 13) to point out the limitation and also emphasize the significance. "Only ~2% of FIPRESCI data are contributed by eRNA reads and there may be many eRNAs are too short to be detected. However, thousands of pairs of eRNA and protein-coding genes can be observed co-expressed in the same single cell, which may provide a unique opportunity to explore gene regulation."

R2 Figure 1

5. The ratio of UMIs for rRNA is surprisingly low for Random as compared to Mix and PolyT. The authors suggest that this feature may be due to rRNA inhibiting Tn5 tagmentation. Is there supporting evidence or literature to claim this? Further, rRNAs are abundantly expressed than any other protein coding genes so the ratio of UMI should also be relatively high. Have authors compared the percentage of detected promoter types, where it should better reflect the diversity of genes that this method can detect (instead of gene expression ratios). Many of lncRNAs are polyA minus but this analysis seems to suggest that lncRNA detection is equal across all conditions. Which lncRNA annotation was used for this analysis? It will be useful to include the analysis results in the revised manuscript.

Answer: Thanks for the reviewer's comment. The gtf file which includes lncRNA annotation is downloaded from https://ftp.ncbi.nlm.nih.gov/genomes/refseq/vertebrate_mammalian/Homo_sapiens/annotation_releases/109/GCF_000001405.38_GRCh38.p12/GCF_000001405.38_GRCh38.p12_genomic.gtf.gz. Our bioinformatics pipeline mainly follows the one used in VASA-seq (Salmen *et al.*, 2022). And we have validated it by using data from VASA-seq to reproduce their Extended Data Fig. 2b. We noticed that whether including multiple mapping reads or not will affect the ratio of rRNA reads. For reads that can be mapped to multiple regions, we randomly chose a hit to report. Then we updated the calculation by including all the mapped reads not only the unique mapped reads (R2 Figure 2a). Of note, the length of the yellow bar does not necessary indicate

“lncRNA detection is equal across all conditions”, but just the relative proportion is similar.

Next, as suggested by the reviewer, we have done a comprehensive literature investigation about scRNA-seq protocols which can detect non-polyadenylated RNAs. Interestingly, we found SUPeR-seq (Fan *et al.*, 2015) is a random primer-based RT one-tube protocol without rRNA depletion. In the main text it mentioned that “Unexpectedly, SUPeR-seq showed no significant amplification of rRNAs, the major RNA species in a cell. No more than 1.5 % of the total SUPeR-seq reads were mapped to rRNAs (Rn5s, Rn5.8s, Rn18s, and Rn28s) when starting with a single cell or single-cell amount of total RNAs”. Since SUPeR-seq is a one-tube protocol, it used a cell lysis buffer that lack Proteinase K. This is similar to FIPRESCI but different from other random primer-based RT protocols. To test whether Proteinase K is relevant, we have performed a bulk version of FIPRESCI RNA-seq experiment. The results suggest when adding Proteinase K to the lysis buffer, the proportion of rRNA reads will significantly increase (R2 Figure 2b). We conclude that the low rRNA ratio of FIPRESCI and SUPeR-seq is at least partly due to the cell lysis buffer without Proteinase K.

As suggested by the reviewer, we have tried to plot the diversity of genes based on promoter types. We merged TSSs which are close to each other less than 500 bp as a single promoter region. For a promoter to be called as detected, we require at least 5 reads. As the total promoter numbers of rRNA and protein coding genes are 8 and 28340 respectively, the % of rRNA promoters detected will be a very small number. Thus, we asked how many percent of promoters are detected for each RNA category. The total promoter number of each category of RNA is indicated in the legend and the height of the bar is the proportion of promoters detected for each category of RNA (R2 Figure 2b).

We have added R2_Figure 2a, b, c as Supplementary Fig. S5.d, e, f and briefly mentioned in the main text (Page 6, Line 28)

R2 Figure 2

6. The violin plot illustrates a strange pattern for fixed condition where the gene detection and UMI are bimodally distributed as compared to fresh condition. This seems to suggest that there is a group of genes that are more amendable to fixation conditions (e.g. possibly due to RNA structure, bound proteins, transcript length and GC content <https://bmcbgenomics.biomedcentral.com/articles/10.1186/s12864-021-07744-6>) than others and cause biases in gene detection and expression levels due to fixation method. This can lead to spurious read counts – especially at the 5' end - thus confounding the conclusions made in this paper. Resolving this issue, e.g. what is the cause of this bias and addressing them will be absolutely critical to evaluate the conclusions made in this manuscript.

Answer: Thanks for the reviewer's comment. As suggested by the reviewer, we have identified top differentially-expressed genes between 10X Genomics 5'-end scRNA-seq using fresh cells and methanol fixed cells (R2 Figure 3a). The fixed specific detected genes are within a normal GC content while fresh specific genes have an unusual high GC content (R2 Figure 3b). There length of those two group genes is different but not significant (R2 Figure 3c). This is partially consistent with Angela R Wu's observation (Wang *et al.*, 2021). The TSO oligos used in 10X Genomics and Smart-seq2 mentioned in (Wang *et al.*, 2021) are different, and different TSOs are known also contribute to the difference among transcriptome detected (Hagemann-Jensen *et al.*, 2020; Jia *et al.*, 2022; Hagemann-Jensen *et al.*, 2022). It will be hard to dissect exactly which component contributes to the difference. On the other hand, it is generally accepted that compare to reverse transcription based RNA-seq, results from

smFISH will be more close to the ground truth of gene expression level (Grün *et al.*, 2014). Although scRNA-seq with the fresh cell is the most popular procedure, we cannot rule out that the difference observed in (R2 Figure 3a) is due to some bias introduced by fresh cell procedure. Because the RNA degradation is not completely stopped even RNase inhibitors are present after the cell lysis. We agree with the reviewer that deepening understanding of the “bias” or “technique variation” is important, but we believe the experiment needs to do is systematically compare sc(n)RNA-seq with the fresh condition, fixed condition, and smFISH. However, this kind of work is beyond the scope of this study.

R2 figure 3

Furthermore, although FIPRESCI is based on the 10X Genomics platform, it is conceptually more similar to sci-RNA-seq or SPLiT-seq, and fixation of the cell is commonly used in those types of protocols. Thus, FIPRESCI should be considered as a SPLiT-seq-like new technique when compared with the standard 10X Genomics procedure. In practice, people usually do not use SPLiT-seq to profiling a sample of

health conditions and then compare it to a sample of disease conditions profiled by 10X Genomics to identify differentially-expressed genes. The solution for preventing making a conclusion based on technical bias is to do the comparison within the dataset generated from the same technique. For example, Cole Trapnell applied sci-Plex(sci- RNA-seq3 with cellular hashing) to screen three cancer cell lines exposed to 188 compounds(Srivatsan *et al.*, 2020). It should be reasonable to expect that most of the differentially-expressed genes identified among different treatments are caused by the drug. It should also be reasonable to expect that the cell type marker identified by sci-RNA-seq or SPLiT-seq is consistent if the same sample is profiled by 10X Genomics. However, if compare the dataset of sci-RNA-seq or SPLiT-seq with 10X Genomics from similar tissue, it is not surprising a lot of differentially-expressed genes will be found. Thus, it is reasonable to conclude the differential expressed genes in Supplementary Figure S10 identified between healthy donor and cancer patients are reliable, because all the data involved in comparison are generated from FIPRESCI protocols.

On the other hand, thanks to the advances in the computational algorithm, the single-cell dataset from different batches, different protocols, and even different modalities now can be integrated with high accuracy. For example, canonical correlation analysis (CCA) based Seurat V3(Butler *et al.*, 2018), integrative non-negative matrix factorization (iNMF) based LIGER(Welch *et al.*, 2019), and neural network based scJoint(Lin *et al.*, 2022) all can integrate atlas-scale, heterogeneous collections of scRNA-seq and scATAC-seq data. J. C. Marioni and colleagues have successfully integrated seqFISH data, which only measures 387 genes, with a 10X Genomics scRNA-seq dataset (Lohoff *et al.*, 2022). ScRNA-seq and snRNA-seq data are believed to be very different. However, the Jay Shendure group has integrated a sci-RNA-seq(nuclei) and 10X Genomics(cell) dataset successfully (Qiu *et al.*, 2022). Jens C. Brüning's group did a comprehensive evaluation of the integration algorithm for scRNA-seq and snRNA-seq data from the hypothalamus, and conclude that scVI performs best (Steuernagel *et al.*, 2022). Theoretically, the integration of data from FIPRESCI and standard 10X genomics procedure is much less challenging than the task mentioned above. Indeed, using the dataset mentioned in the previous response, we have evaluated Harmony, Seurat V3(CCA), and scVI and found Seurat V3(CCA) can remove the technique variation and batch effects between the two datasets pretty well even without tuning the parameters (R2 Figure 3d, e). The difference between our analysis and Jens C. Brüning's work may be due to we didn't systematically optimize the parameters for scVI. Thus, the comparison between FIPRESCI and publicly available 10X Genomics data in Figure 3b and Supplementary Figure S9c, which get similar cellular composition results from a similar type of sample, further support that our approach and results are reliable.

To address the reviewer's concern and take note of the limitation of FIPRESCI, we have added the statement below to the discussion section (Page 11, Line 33) and cited Angela R Wu's BMC Genomics paper mentioned by the reviewer. "Similar to the previous report, we found methanol fixation does not affect the cell type clustering, but certain types of biological analyses that may be influenced by the GC-

content or transcript length need to use caution when interpreting the results from FIPRESCI data with methanol fixation”.

7. Although protein expression imputation would be useful for the reasons mentioned by the authors, it can be implemented on all single cell RNA-seq datasets in general, and the authors have not shown any particular advantage in performing this analysis with the FIPRESCI dataset. The motivation for protein expression imputation and these results should be placed in the discussion section as a potential application for FIPRESCI.

Answer: Thanks for the reviewer’s comment. We agree with the reviewer and have moved Fig. 4b to Supplementary Fig S9a, and mentioned this part in the discussion section (Page 11, Line 18).

Additional comments:

While the detection of eRNAs under the pretext that they are within ATAC peaks is notable, there are ample spurious reads that are mapped across the human genome derived from single cell RNA-seq methods including 10x 5’. Therefore, it will be imperative for authors to demonstrate that these eRNAs are significantly more enriched than reads that are outside of ATAC peaks, plus the drop off rates based on varying degree of threshold set for RPM as 0.5 is arbitrary and difficult to grasp the sensitivity and accuracy of this level. The response from authors states that on page 20 line 28 provides more detail statistics and methods but the revised version lacks description and should provide a more detailed statistics the claim that these eRNAs are in fact true and not derived from random/spurious mapped reads. Furthermore, eRNAs exhibit bidirectional transcription. Please demonstrate that these eRNAs are divergently transcribed at the single base resolution derived from TSS (e.g. in Figure 2d) and that they largely overlap with canonical histone markers including H3K27ac.

Answer: Thanks for the reviewer’s comment. To evaluate the baseline level of FIPRESCI reads and exclude the signal from protein coding genes and other RNA from regulatory elements, we have calculated the RPKM from H3K9me3(GSE198978) peaks (R2 Figure 4a bottom). In contrast, the FIPRESCI signal from distal ATAC peaks has a very different distribution (R2 Figure 4a upper). The result clearly shows that threshold of RPKM>1 should separate eRNA signal from random noise. In addition, we require the core region have significant (adjusted p-value <0.01, Wilcox test and Benjamini-Hochberg p-value adjustment) higher signal than the surrounding region. See more detail in (Page 21, Line 4) in the revised manuscript. Based on this result, we have updated Figure 2e,f,g.

We noticed that in our FIPRESCI data, only the read2 from a paired-end reads are exactly start from the 5’-end of the RNA. The read1 are from the internal of the RNA, and using the mixture of read1 and read2 to plot Figure 2d will lose the single base resolution for identifying enhancer TSSs. Thus, we have re plotted the Figure 2d with only read2. There is a bimodal distribution centered around the start and end site of the distal ATAC peak. Those results support that eRNAs we identified are bidirectional transcribed.

For the updated set of eRNAs, we compared them with a public available H3K27ac

data (GSM733684). The result suggests that more than 60% of the eRNA are overlapped with H3K27ac peaks (R2 Figure 4b). And across the condition, the eRNA result is highly correlated. Since the requirement of both ATAC peaks and high FIPRESCI signal is stringent, we speculate the eRNA candidates in our results are from strong enhancers. And eRNA from weak enhancers may be missed in our analysis.

R2 Figure 4

Reference:

- Butler, A. *et al.* (2018) Integrating single-cell transcriptomic data across different conditions, technologies, and species. *Nat. Biotechnol.*, **36**, 411–420.
- Fan, X. *et al.* (2015) Single-cell RNA-seq transcriptome analysis of linear and circular RNAs in mouse preimplantation embryos. *Genome Biol.*, **16**, 148.
- Grün, D. *et al.* (2014) Validation of noise models for single-cell transcriptomics. *Nat. Methods*, **11**, 637–640.
- Hagemann-Jensen, M. *et al.* (2022) Scalable single-cell RNA sequencing from full transcripts with Smart-seq3xpress. *Nat. Biotechnol.*
- Hagemann-Jensen, M. *et al.* (2020) Single-cell RNA counting at allele and isoform resolution using Smart-seq3. *Nat. Biotechnol.*, **38**, 708–714.
- Jia, E. *et al.* (2022) Correction to: Optimization of library preparation based on SMART for ultralow RNA-seq in mice brain tissues (BMC Genomics, (2021), 22, 1, (809), 10.1186/s12864-021-08132-w). *BMC Genomics*, **23**, 1–15.
- Lin, Y. *et al.* (2022) scJoint integrates atlas-scale single-cell RNA-seq and ATAC-seq data with transfer learning. *Nat. Biotechnol.*
- Lohoff, T. *et al.* (2022) Integration of spatial and single-cell transcriptomic data elucidates mouse organogenesis. *Nat. Biotechnol.*, **40**, 74–85.
- Qiu, C. *et al.* (2022) Systematic reconstruction of cellular trajectories across mouse embryogenesis. *Nat. Genet.*, **54**, 328–341.
- Salmen, F. *et al.* (2022) High-throughput total RNA sequencing in single cells using

- VASA-seq. *Nat. Biotechnol.*
- Srivatsan, S.R. *et al.* (2020) Massively multiplex chemical transcriptomics at single-cell resolution. *Science (80-.)*, **367**, 45–51.
- Steuernagel, L. *et al.* (2022) HypoMap—a unified single-cell gene expression atlas of the murine hypothalamus. *Nat. Metab.*, **4**, 1402–1419.
- Wang, X. *et al.* (2021) The effect of methanol fixation on single-cell RNA sequencing data. *BMC Genomics*, **22**, 1–16.
- Welch, J.D. *et al.* (2019) Single-Cell Multi-omic Integration Compares and Contrasts Features of Brain Cell Identity. *Cell*, **177**, 1873-1887.e17.